# Carbon losses from deforestation and widespread degradation offset by extensive growth in African woodlands

Iain M. McNicol [1], Casey M. Ryan [1] & Edward T.A. Mitchard [1]

Land use carbon fluxes are major uncertainties in the global carbon cycle. This is because carbon stocks, and the extent of deforestation, degradation and biomass growth remain poorly resolved, particularly in the densely populated savannas which dominate the tropics. Here we quantify changes in aboveground woody carbon stocks from 2007–2010 in the world's largest savanna—the southern African woodlands. Degradation is widespread, affecting 17.0% of the wooded area, and is the source of 55% of biomass loss ($-0.075$ PgC yr$^{-1}$). Deforestation losses are lower ($-0.038$ PgC yr$^{-1}$), despite deforestation rates being 5× greater than existing estimates. Gross carbon losses are therefore 3–6x higher than previously thought. Biomass gains occurred in 48% of the region and totalled $+0.12$ PgC yr$^{-1}$. Region-wide stocks are therefore stable at ~5.5 PgC. We show that land cover in African woodlands is highly dynamic with globally high rates of degradation and deforestation, but also extensive regrowth.

---

[1] School of Geosciences, University of Edinburgh, Edinburgh EH9 3FF, UK. These authors contributed equally: Iain M. McNicol, Casey M. Ryan. Correspondence and requests for materials should be addressed to I.M.M. (email: i.m.mcnicol7@gmail.com)

Carbon fluxes from vegetation growth and land-use change are major uncertainties in the global carbon cycle[1,2]. Deforestation and degradation are reducing woody carbon stocks[1,3,4], although tree growth and woody expansion may be counterbalancing these losses[5]. The location and magnitude of these changes are poorly resolved[6], particularly in the densely populated savannas and woodlands which dominate the tropical land surface[2]. These seasonally dry ecosystems, which are characterised by an open tree canopy and a continuous grass layer, are the dominant vegetation of Southern Africa. Human activities are thought to be driving widespread and rapid changes in woody cover across the region, with potentially important implications for both the global carbon cycle[7–11], and local livelihoods, as over 150 million people depend on ecosystem services provided by the woodlands and forests. Rising populations, stagnant yields, altered consumption preferences and new connections to the global economy are thought to be driving widespread deforestation (a reduction in wooded area), mostly due to agricultural expansion, and degradation (a reduction in woody carbon density in an area that remains woodland), often due to harvesting timber or fuel wood[4,7,12]. However at the same time, several processes are hypothesised to be increasing woody carbon stocks in the region, including widespread and rapid regrowth following shifting cultivation[13], enhanced tree growth stimulated by increased atmospheric $CO_2$ concentrations[14,15] and reductions in browsing megaherbivores[16]. Yet, the location and rates of these processes, particularly the extent of woody degradation, biomass growth and regrowth, and the impact of these changes on aboveground woody carbon stocks (AGC), are largely unknown[2].

Addressing these uncertainties is hampered by a lack of knowledge of the carbon-area (MgC ha$^{-1}$) density of the woodlands, and its changes over time. Existing coarse resolution maps of AGC have large discrepacies over African woodlands[17–19], whilst the seasonality[20] and mixed tree-grass structure of savanna woodlands challenges optical remote sensing estimates of tree cover change[4,21], as green leaf area and reflectance are dynamic and weakly linked to woody biomass. In addition, many of the current datasets on woodland dynamics, including the UN Food and Agricultural Organisation's (FAO) Forest Resource Assessment (FAO-FRA)[22], present an incomplete picture of woodland and forest dynamics by failing to account for the low intensity, but widespread, losses occurring in these systems due to wood harvesting, fire and selective logging[7,10,23] (i.e. degradation), or the extent of AGC gains, which are largely unmeasured[11]. Obtaining both accurate and spatially explicit estimates of deforestation, degradation, and biomass (re)growth are crucial to evaluate the response of savanna woodlands to global change[24], and also to support accurate resource assessment, land management, and the effective targeting and monitoring of land use emission abatement policies.

Here we generate novel estimates of the rates, locations and carbon stock changes associated with degradation and biomass (re)growth across Southern African woodlands, and provide new, contrasting data on deforestation. These estimates are derived from 25 m resolution maps of AGC across Southern African for 2007–2010, created using a combination of space-borne L-band radar imagery and field data. A key advantage of using radar data is that unlike optical imagery, it is largely insensitive to the intra- and inter-annual variability in the grass layer and tree leaf phenology[25], which can hinder accurate change detection. Instead, L-band backscatter from ALOS PALSAR[26] is known to strongly correlate with woody biomass at multiple sites across African savanna woodlands[27–29], where it has been used to detect small scale changes associated with shifting cultivation and tree harvesting[8,12], as well as areas of increasing biomass at larger scales[30]. In this paper, we extend these analyses across the full extent of the Southern African savanna woodlands and dry forests. We find that degradation is widespread and the principal source of carbon loss across the study region. Carbon losses via deforestation are around half of those resulting from degradation. As such, gross carbon losses are greater than previously thought; yet total aboveground carbon stocks are relatively stable over time due to extensive biomass gains, largely in remote areas.

## Results

**Woody cover and aboveground carbon stocks**. Our study region includes all southern African countries where savanna woodlands are the dominant vegetation type (Supplementary Fig. 1), including Angola, Zambia, Zimbabwe, Malawi, Tanzania, Mozambique, and the southern parts of the Democratic Republic of Congo (formerly Katanga Province)[31,32] (Fig. 1). The area of woodland and forest, defined here as pixels with an AGC density ≥10 MgC ha$^{-1}$ in 2007, was 2.3 M km$^2$ (95% confidence intervals (CI): 2.1–2.5 M km$^2$, based on the uncertainty of the biomass–backscatter relationship—see Methods). Our estimate is similar to the 2005 forest area estimate from the FAO FRA (Fig. 1) and equates to 50% of the total land area (Supplementary Data 1), and 10% of the estimated global tropical forested area[33]. The estimates of tree cover from Hansen et al.[4] (hereafter Hansen) are markedly higher than our data, and that of the FAO-FRA, even after applying a similar forest definition (10% tree cover) (Fig. 1). Nationally, wooded area varies from 33% in Zimbabwe to 55% in Mozambique and Zambia and 62% in the (former) Katanga province of the Democratic Republic of Congo.

The mean region-wide carbon density was 24.0 [19.8–28.5] MgC ha$^{-1}$, with the most carbon dense woodlands located in Katanga (mean 28.6 MgC ha$^{-1}$), and the least dense in Zimbabwe (19.6 MgC ha$^{-1}$; Fig. 2). Total AGC stocks were approximately constant from 2007 to 2010 (Fig. 2; Supplementary Data 1), being estimated at 5.51 [4.90–6.14] PgC for 2007 and 5.46 [4.8–6.10] PgC in 2010, equivalent to 2–3% of the tropical biomass stock[1,18,33], with similar values in 2008 (5.41 PgC) and 2009 (5.42 PgC) (Supplementary Fig. 2). Carbon stored in other wooded lands, defined as areas with an AGC density <10 MgC ha$^{-1}$ in 2007, totalled ~0.7 PgC, however, owing to the higher uncertainty in change detection in low AGC areas, we do not consider the land cover change dynamics of these areas (see Methods).

Our AGC estimates (excluding Katanga) are 61% lower than the FAO FRA[22], and 66 and 33% lower than the pan-tropical AGC maps of Baccini et al.[3] and Saatchi et al.[33] respectively. The recent fusion of multiple datasets by Avitabile et al.[18] yielded total AGC estimates only +3% higher than our own, [−9 – +12%], albeit with large spatial differences between the two datasets, with Avitabile et al. estimating considerably higher stocks in high AGC areas relative to our data, alongside lower estimates in lower density areas (Supplementary Fig. 3).

**Land-cover change**. The aggregate temporal stability in regional carbon stocks between 2007 and 2010 conceals the presence of widespread gross gains and losses, and large differences between the east and west of the region (Fig. 3). Biomass gains were detected in 48% [41–55%] of the wooded area, with deforestation and degradation occurring in 8.4% [6.4–9.9%] and 17.0% [14.0–19.7%], respectively (see Methods for a definition of these land cover changes).

Degradation rates were highest in Mozambique, Malawi and Tanzania (Fig. 4), with hotspots located near large, rapidly expanding urban centres where woodland resources are scarce (e.g. Dar es Salaam, Luanda and southern Malawi)[7,34], and along transport corridors, ports and some borders (e.g. Beira, Nacala

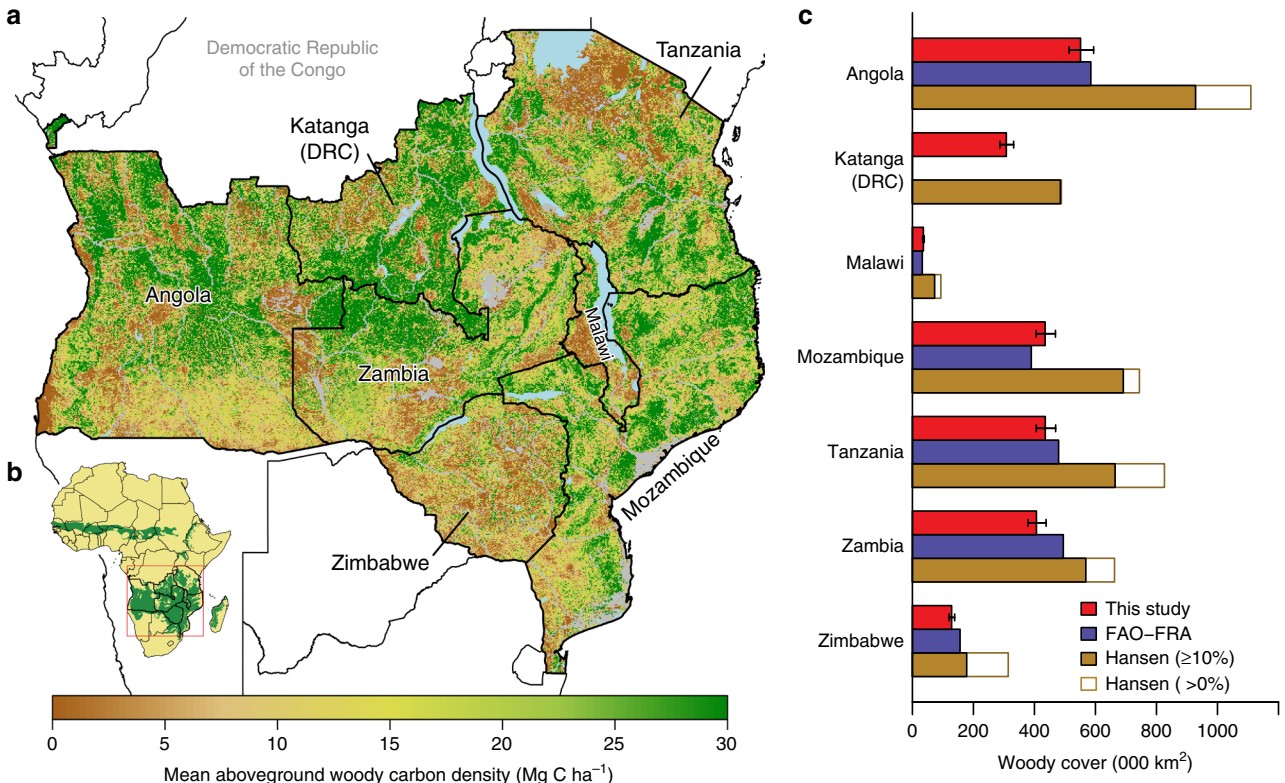

**Fig. 1** Spatial distribution of aboveground woody carbon stocks in 2007, estimated from L-band radar data and in situ measurements. **a** The mean carbon density, averaged to 1 km² for display purposes, with non-wooded areas (<10 MgC ha⁻¹) and other areas masked from the analysis in grey. **b** The location of our study region. **c** Wooded ('forest') area in each country according to our study, the FAO-FRA (2015) and the Hansen et al. (2013) dataset

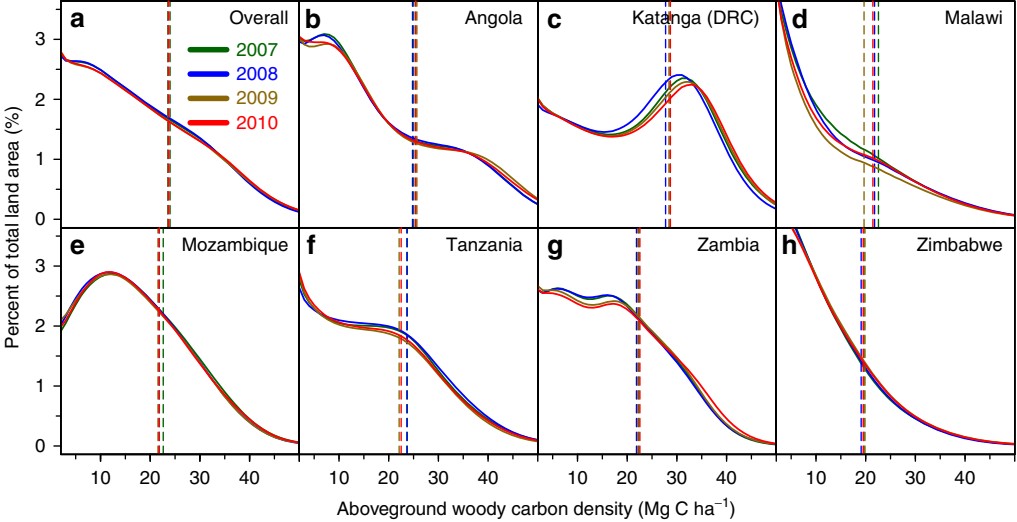

**Fig. 2** Frequency distribution of aboveground woody carbon densitites. **a–h** The distribution of aboveground woody carbon densities, aggregated to 1 ha resolution, for our study region (**a**), and for each country (**b–h**), including the woodland dominated former Katanga Province of the DRC, from 2007 to 2010. The vertical lines indicate the yearly mean

and southern Tanzania)[8,12]. The spatial pattern (Supplementary Fig. 4) suggests these hotspots might be linked to urban demand for biomass energy, and domestic and international timber markets, as opposed to the subsistence needs of the local population[7,8,12]. Degradation typically reduced AGC from 29 ± 10 MgC ha⁻¹ (mean ± SD) to 20 ± 4 MgC ha⁻¹, with degradation disproportionately prevalent in higher biomass woodlands (Fig. 5). This suggests these areas are being targeted for

harvesting, probably because they contain trees of suitable size and species for charcoal and timber.

The area affected by deforestation (193,000 [158,000–214,000 km²]) was around half (49%) the area degraded, with deforestation rates ranging from 1.8% yr⁻¹ in Katanga to 4.7% yr⁻¹ in Malawi, and exceeding 2% yr⁻¹ in the remaining countries (Fig. 4b; Supplementary Data 1). In contrast to degradation, the vast majority (95%) of deforestation was located in areas with a

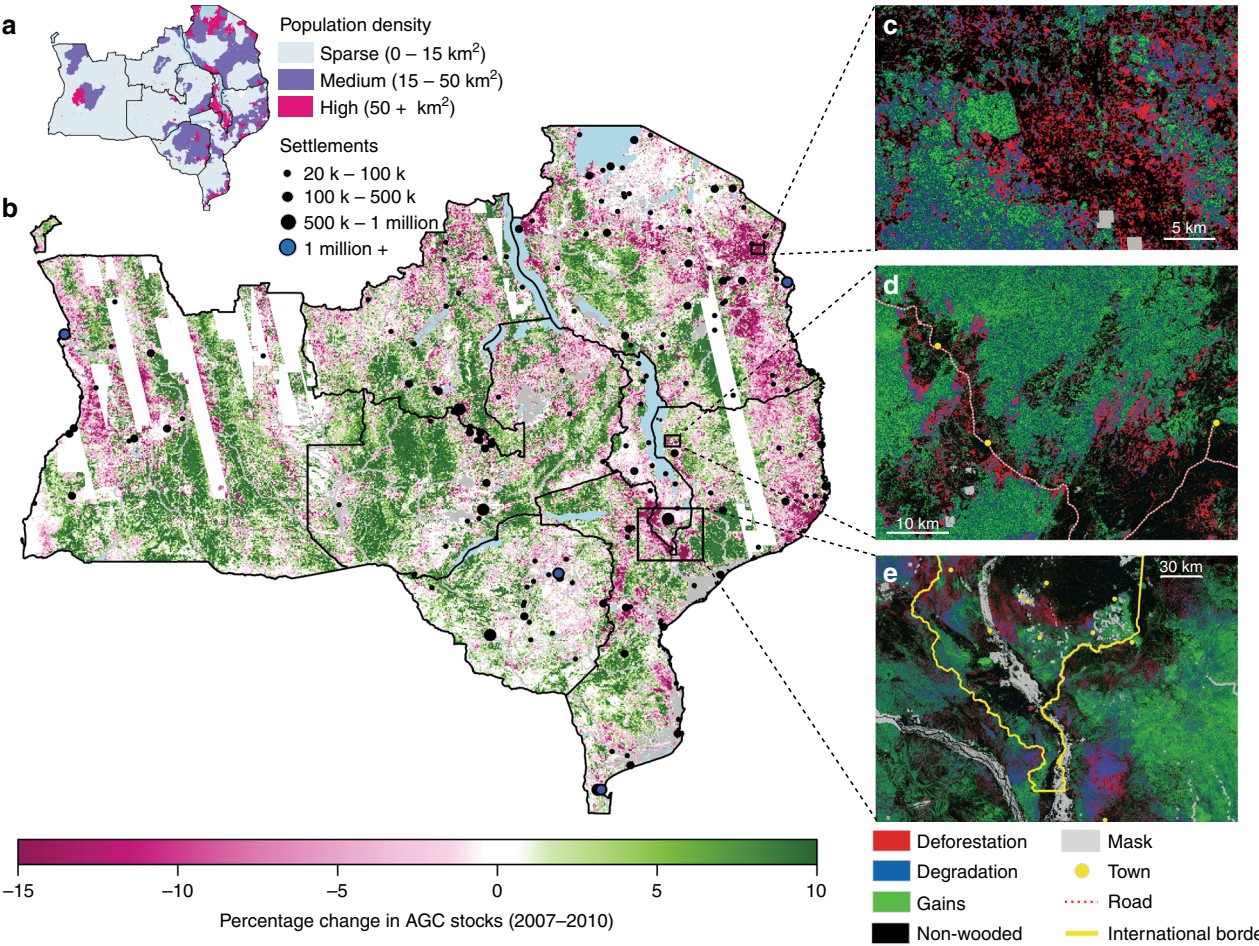

**Fig. 3** Change in above ground carbon stocks between 2007 and 2010. **a** The human population density and centres across the study region[58]. **b** The mean percentage change in aboveground woody carbon stocks (AGC) at 1 km resolution. Areas masked from the study due to soil moisture differences between years are masked by the white stripes, whilst irrigation or urban land covers are masked in grey. The sub figures **c–e** are at 25 m resolution and illustrate three important syndromes of land cover change: **c** the deforestation (red) of small areas of woodland in an already largely deforested area of Tanzania (mopping up), **d** the progression of the agricultural frontier and associated deforestation and degradation (blue) to the north of Lichinga city in Mozambique, and **e** the extensive degradation in frontier regions of Mozambique near to the demand centres of southern Malawi, suggestive of cross border flows of biomass energy

lower than average AGC density (mean biomass change: $14 \pm 4$ MgC ha$^{-1}$ to $6 \pm 5$ MgC ha$^{-1}$; Fig. 5; Supplementary Fig. 4)[8], being particularly prevalent in already fragmented agricultural landscapes, as typified in Fig. 3c, as opposed to the frontier-style deforestation characterised in Fig. 3d.

The total area of deforestation estimated here is 4.6 and $5\times$ higher than previous estimates by both the FAO-FRA[22] (pro rata for 2005–2010, excluding Katanga) and Hansen et al.[4] for the same time period (2007–2010) respectively. This larger deforested area was observed despite our smaller starting estimate of wooded area (due to a stricter forest definition; Fig. 1), meaning our overall percentage deforestation rate (2.8% yr$^{-1}$) is 7.8 and $9.4\times$ higher than the estimates from FAO and Hansen (Fig. 4b). Increasing the tree cover threshold used to calculate 'forest' area in Hansen dataset to 10% resulted in only a small increase in the estimated rate of deforestation, meaning our conclusions are robust. These contrasting rates and area estimates are in part due to the differing definitions of deforestation, with Hansen mapping areas of complete tree cover loss, whilst the FAO statistics are based on extrapolated rates of change using diverse land use classifications. The FAO estimates are also a net change figure which includes estimates of both forest loss and expansion

meaning that the gross area and rate of loss are likely to be higher. In contrast, our approach allows for the presence of residual trees in deforested areas[8,13], with only 10% of our deforested area comprising areas that were completely cleared. We detect deforestation in 59% of the locations where Hansen find deforestation, and observe degradation in a further 21%, with the remainder of the Hansen deforested area (20%) almost fully accounted for by areas masked from our analysis, or not considered woodland in 2007 (Supplementary Table 2). In contrast, Hansen observed deforestation in only 19% of our deforested areas, increasing to 38% when only areas of complete clearance (i.e. 0 MgC ha$^{-1}$) are considered, indicating that differing definitions of deforestation only partially explain the differences. Most of our extra deforestation occurred in areas with low biomass and low tree cover in 2007 (Supplementary Fig. 5 and 6) suggesting our method is more sensitive to changes in sparsely wooded areas. In such areas, crop or grass biomass strongly influence the optical signal which could lead to deforestation remaining undetected in the Hansen product if the removal of trees only weakly affects the land surface reflectance.

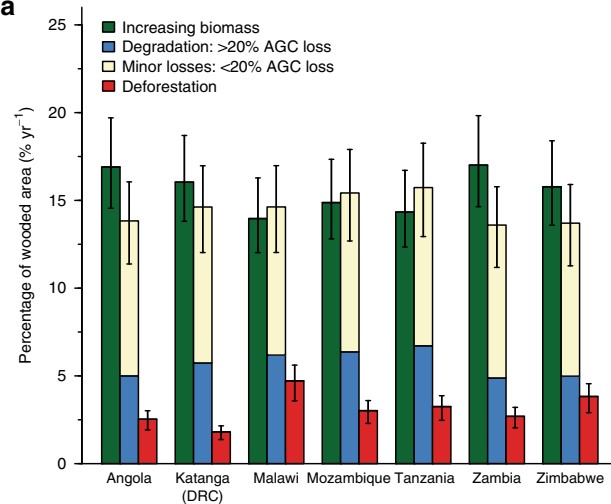

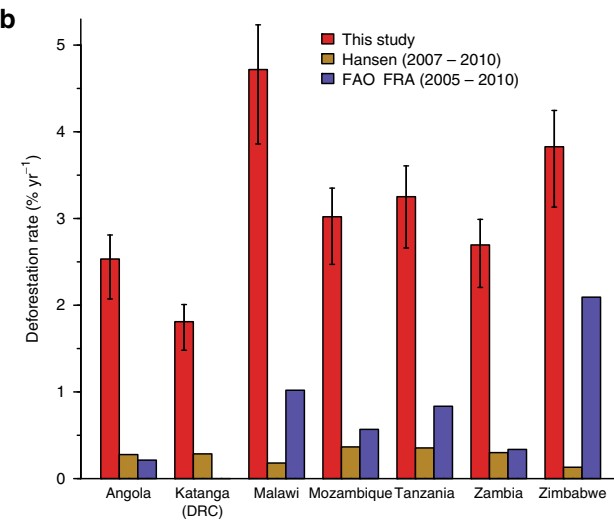

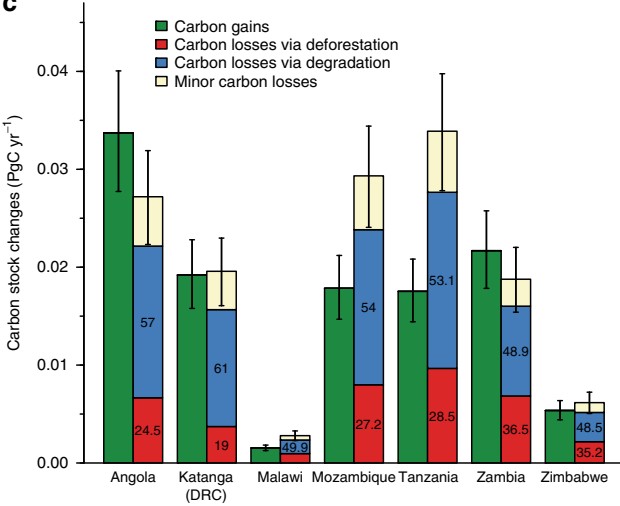

**Fig. 4** Woodland area and carbon stock changes separated by nation/region. **a** Percentage of the wooded area in 2007 affected by deforestation, degradation, minor losses and (re)growth. **b** Comparison between existing estimates of deforestation rates and this study. **c** Carbon stock changes due to deforestation, degradation and (re)growth, with the values is the losses bar showing the percentage contribution of deforestation and degradation to the total carbon losses . Error bars show the 95% confidence intervals (CIs) and represent for the total error on each bar

Alongside these rapid losses, we also find evidence of widespread gains in biomass ($1.3 \pm 0.9$ MgC ha$^{-1}$yr$^{-1}$; median $\pm$ SD), the magnitude of which is consistent with field data on regrowth rates[13]. Biomass increases were more prevalent within relatively low biomass stands, with 60% of the total gain area in woodlands with AGC < 25 MgC ha$^{-1}$ in 2007 (Fig. 5). Widespread gains were also observed in areas that are sparsely populated, and/or have a relatively high mean annual precipitation, including the southern and western parts of Tanzania and Angola, western Zambia and southern DRC (Fig. 3)[9,34,35]. Extensive gains are to be expected given the ubiquity of disturbance in this ecosystem and the typically rapid subsequent regrowth[13]. However, our finding that regional AGC stocks are roughly constant, despite 24% of the region being deforested or degraded over the 3-year period, implies some non-equilibrium (re)growth, which may be caused by enhanced disturbance rates prior to the study period (e.g. due to more severe fire regimes in a less fragmented landscape[10], or higher elephant densities), possibly combined with enhanced current growth rates (which are predicted under increased $CO_2$[14,15]).

**Carbon stock changes**. Carbon stock changes (which describe the carbon committed to the atmosphere, a proxy for emissions) associated with deforestation, degradation and (re)growth were estimated by weighting the observed carbon stock changes by the probability of each land cover change having occurred in each pixel. Over the 3-year period, biomass changes due to (re)growth were 0.35 [0.29–0.42] PgC with losses totalling 0.41 [0.34–0.48] PgC, of which deforestation contributed 0.11 [0.08–0.14] PgC, and degradation 0.22 [0.17–0.27] PgC, with the remainder being minor losses which can often be due to natural processes. Thus degradation accounts for 66% [59–72%] of the likely anthropogenic (deforestation + degradation) carbon losses (Fig. 4c) and 55% [41–68%] of total gross losses. There were large variations in carbon dynamics across the region, with net reductions in Mozambique (−3.4% of 2007 AGC stock), Tanzania (−4.8%) and Malawi (−4.9%), and gains in Angola (+1.5%) and Zambia (+1.0%) (Fig. 4c).

The small, insignificant net change in AGC observed here (−0.02 [−0.4–0.4] PgC yr$^{-1}$) contrasts with FAO-FRA[22] statistics which suggest a more rapid reduction in stocks across the study region (−0.08 Pg yr$^{-1}$; 2005–2010), probably because our dataset better accounts for biomass gains (Supplementary Table 3). Our estimate of gross AGC losses from deforestation and degradation (0.11 PgC yr$^{-1}$ [0.09–0.14] PgC yr$^{-1}$) is 6× higher than that obtained from overlaying the most widely used maps of forest change (Hansen[4]) and carbon stocks (Avitabile[18]) (0.016 PgC yr$^{-1}$), and 3× higher than the recent estimate by Baccini et al.[23] These boosted carbon emissions primarily reflect the incorporation of degradation losses, but also the higher deforestation area, and differences in the carbon density of land undergoing change. Our estimated gross anthropogenic AGC losses are similar to those released from deforestation in the more spatially extensive Brazilian Amazon (0.18 Pg yr$^{-1}$)[36], and are equivalent to 4–10% of the current estimated gross tropical land-use emissions[3,5,6], indicating the importance of these woodlands for the global climate system.

## Discussion

Overall, our results present a picture of highly dynamic land cover change across the region, with rapid deforestation and degradation underway in hotspots around population centres. Carbon losses from deforestation and degradation are markedly higher than the current best-guess estimates for our study region[1,4,18,23], yet these losses are offset by previously undetected,

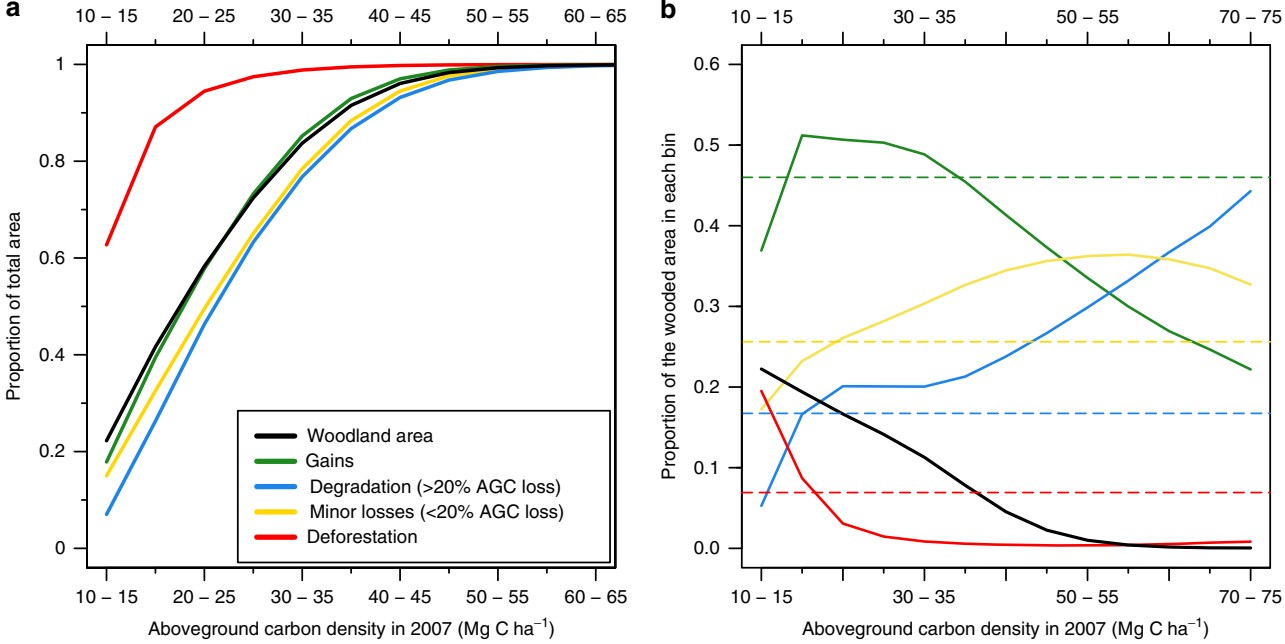

**Fig. 5** Area of land cover change by the initial carbon density. **a** The cumulative contribution of different AGC classes (in 5 MgC ha$^{-1}$ bins) to the total area of woodland, and each land cover change. **b** The proportion of woodland areas in 2007 that were affected by each land cover change. The hatched horizontal lines indicate the proportion of the total woodland area affected

but widespread gains in carbon stocks, which occur mostly in more remote areas. Our results contrast with the recent study by Baccini[23] which suggests that the study region is a net source of carbon emissions. In this analysis we exclude areas that were non-wooded in 2007, which precludes the estimation of non-forest carbon gains and losses, including wooded area expansion, which could also be widespread[9]. Our results have clear implications for the global carbon cycle, particularly if the scale of this under-estimation in both carbon losses and gains is replicated across the seasonally dry tropics.

Degradation, which has never been quantified at this scale in a spatially explicit manner, is the main cause of biomass loss[23,37], being particularly prevalent in higher biomass areas, which are often floristically diverse and of high conservation value[38]. African savanna woodlands are unique in that they retain significant wooded area and biomass, alongside a high human population closely dependant on woodland resources. Thus, large degradation losses may be a unique feature of African woodlands, but are still large enough to impact global land use emissions.

Our finding of high carbon losses from degradation presents several challenges to attempts to reduce carbon emissions from deforestation and degradation (REDD+). Firstly, most efforts have focussed on avoided deforestation in intact woodlands, whereas we show that degradation, and deforestation in low biomass, mosaic landscapes, are the critical processes. This mis-targeting is potentially costly, as for many regulating and provi-sioning ecosystem services, the last tree to be felled is much more valuable than the first. Secondly, since degradation is difficult to monitor, most REDD + policy and practice has been created with little data on the rates and locations, and thus causes, of degra-dation[12]. This links to a further challenge identified here: the spatial pattern of degradation indicates that it is mostly driven by distal actors, and probably linked to demand for energy and timber in urban areas, or abroad[7,39]. Locally driven, rural demand is unlikely to be a useful intervention point for mitigation, which should instead focus on urban and international value chains and demand.

Our results also highlight the extent of biomass (re)growth across these woodlands, which counterbalance the carbon losses. The dominant miombo and mopane woodlands have long been subjected to, and thus are highly adapted[13] to, disturbance by hominids, fire, elephants and other browsers. Many of these disturbance agents have declined markedly due to urban migra-tion, defaunation, and landscape fragmentation[10], which may explain some of the widespread gains observed in more remote, rural areas. Thus, the disturbances which we find to be wide-spread around population centres may have replaced these quasi-natural losses. The critical issues to maintaining ecosystem service provision and carbon storage is therefore the post disturbance land use, as when woodlands are not fully transformed, they can recover their biomass within three decades of clearance[13], meaning they can support some level of clearing in perpetuity. Continued monitoring of these systems is needed to evaluate the permanence of these land cover changes, and to evaluate the impacts of changes in climate and atmospheric $CO_2$ concentra-tions—drivers which are likely to have contrasting effects on woody cover over the next century[40].

The methods presented here are not specific to the radar satellite used (ALOS PALSAR), and are applicable to longer wavelength radar sensors, including the P-band BIOMASS mis-sion[41] designed to estimate biomass in more carbon dense moist tropical forests, and the planned L-band SAOCOM-1 and NISAR missions[42]. Future monitoring efforts will need to incorporate repeat in situ observations of both AGC growth and loss to corroborate remotely sensed estimates of change—something that is not possible here due to the lack of region-wide con-temporaneous measurements of biomass change. Future work should also focus on understanding the drivers of woodland loss, and the effectiveness of protected areas in reducing land use change.

## Methods

**Study area**. Our study region includes all southern African countries where savanna woodlands - mixed tree-grass ecosystems consisting of an open tree layer and a continuous grass layer - are the dominant vegetation type[31,32,43]. This

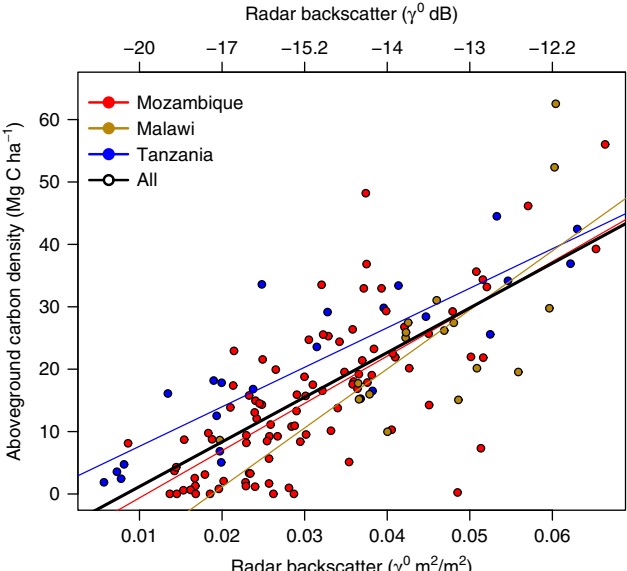

**Fig. 6** Biomass-backscatter relationship. HV radar backscatter ($\gamma^0$) plotted against aboveground woody carbon stocks for 137 sites across the region. Model coefficients and goodness of fit statistics are shown in Supplementary Table 1

includes Angola, Zambia, Zimbabwe, Malawi, Tanzania and Mozambique, and the southern parts of the Democratic Republic of Congo (formerly Katanga Province). The decision to restrict the study to the extent of savanna woodlands stems from the fact that our primary data source (L-band radar) is sensitive to woody carbon stocks up to a saturation point of around 75 MgC ha$^{-1}$. Most savanna woodlands lie beneath this value, meaning the method is able to detect changes in these areas[27,44]. We use political rather than vegetation boundaries to define our study region so as to provide country-specific data on AGC gains and losses that is relevant to policy makers.

The region corresponds to the Zambezian region identified in Linder et al.[45] as being biologically distinct from other African regions in terms of plants, mammals, amphibians, and reptiles, with high levels of diversity and endemism. Although our study region is defined by the presence and dominance of savanna woodlands, variations in climate, soils and disturbance mean that our study region encompasses a structurally and compositionally diverse mosaic of habitats, covering a spectrum from open savanna with a dominant grass layer and scattered trees, through open canopy woodland with an understory of grasses and shrubs, to denser woodlands and dry forest (Supplementary Fig. 1).

The dominant vegetation type is miombo woodland, the vernacular name for tree species in the genera *Brachystegia*, which along with *Julbernardia* is largely endemic to region and helps differentiate these typically open canopy woodlands from other savanna type ecosystems. At the woodier end of the spectrum, woodlands eventually grade in to closed canopy dry forest, such as those in the Eastern Arc Mountains of Tanzania and the East African Coastal Forests. The climate is characterised by distinct wet and dry seasons with 95% of the annual rainfall falling within a period of 3–7 months[35]. Regular dry season fires fuelled by the senescing grass layer are a characteristic feature of these systems and are thought to be a major constraint to woody growth[46]. Human population density is high compared to humid forest regions, and there is a heavy reliance on the woodlands for local livelihoods across the region[40]. Increasing human pressure linked to resource extraction is known to be driving widespread, but uncertain, losses of AGC, as well the localised extinction of important tree species[7,8], the majority of which is likely being driven by subsistence agriculture and the small-scale production of cash crops[12,47]. Shifting cultivation is the traditional method of agriculture in the region[13], with agriculture often the main source of income among local communities. At the continental scale, the shifting cultivation cycle is thought to be a significant source of carbon to the atmosphere[48], although the initial loss is partially, if not fully, offset by the subsequent regrowth[13]. Traditionally, these landscapes have been of little commercial value to farmers due to the relatively infertile soils and the prevalence of Tsetse flies which largely prevent the keeping of cattle. This may be changing with shifting cultivation systems gradually being replaced by more intensive farming practices and a shift to more export-oriented activities[49,50].

**Approach**. The basis of our approach is the creation of a time series of aboveground woody carbon (AGC) density maps, which were generated using a combination of satellite radar images and in situ carbon stock estimates[8,12,27]. The plot

data used for calibration form part of the Socio-Ecological Observatory for the Southern African Woodlands database and are available for use in line with the SEOSAW Code of Conduct (https://seosaw.github.io). These carbon density maps are used to quantify the areal extent of four land cover changes of interest - degradation, deforestation, minor losses and biomass gains - and the carbon stock changes associated with each of these processes. These land cover and biomass change estimates are reported at national and subnational level (Supplementary Data 1). We used the Geospatial Data Abstraction Library (GDAL, 2017), implemented using the Python programming language version 3 (Python Software Foundation, https://www.python.org/), for all of our data processing. All statistical analyses and Figures were created using R Statistical Software (R Core Team, 2014). More detailed descriptions of our approach are available in the Supplementary Methods.

**Radar data**. Radar imagery was obtained from the Phased Array L-Band Synthetic Aperture Radar sensor on-board JAXA's Advanced Land Observation Satellite (ALOS-PALSAR). We use the 25 m horizontal-send vertical-receive (HV) polarisation mosaic product which provides annual maps of radar backscatter for 2007–2010[26] L-band HV backscatter is known to be sensitive to woody biomass density up to a saturation point of around 75 MgC ha$^{-1}$ and has been shown to be able to detect deforestation, degradation, and regrowth in savanna woodlands[8,12,27]. The mosaic product comprises images obtained throughout the year (April—December) and has terrain and radiometric corrections applied. The raw digital numbers were converted to backscatter ($\gamma^0$ in decibels; the ratio of the power returned to the sensor relative to the energy emitted, expressed on the decibel scale) using the calibration coefficients of Shimada et al.[51] after which they were converted to natural units to allow arithmetic, not geometric means to be used in subsequent analyses[8]. Images were filtered using an Improved Lee Filter[52] with a 5 × 5 window to reduce the effect of speckle, a noise-like quality inherent in radar which results from interference among the signal from individual scatterers within a pixel[53].

Estimating aboveground woody carbon stocks. To generate maps of AGC, we regressed backscatter ($\gamma^0$) (in natural units, not dB) against the equivalent field measured carbon stocks at 137 sites in Malawi, Mozambique and Tanzania (median plot size 0.6 ha) (Fig. 6). We derived a general model (AGC = 715.67 × $\gamma^0$–5.97; $r^2$ = 0.57; cross validation RMSE = 8.5 MgC ha$^{-1}$; bias = 1.1 MgC ha$^{-1}$), which we use to convert radar backscatter to maps of AGC (See Supplementary Methods). The RMSE represents the error on a prediction of biomass for a single pixel, which decreases as pixels are aggregated together (i.e. RMSE is minimal at the scale of the districts that we report here). However, bias is a separate quantity from RMSE and does not cancel out. As such, bias is the main source of uncertainty in biomass estimation at regional scales[8]. Our method is built on the commonly applied assumption that there is a relationship between the remotely sensed quantity and the biophysical attribute of the land surface which holds true over time[8,11,34,54], and that this is valid for the range of vegetation types in our study area. The assumption of an invariant physical relationship between the remotely sensed observation and land surface is common among change detection studies using radar and other sensors[8,11,34,55]. In our case soil moisture effects have been accounted for (see below), and the stability of the sensor response over time has been verified[26], meaning any changes in backscatter are likely to be related to changes in land cover. A more detailed description of the model fitting procedure along with additional information on the field plots used for calibration is located in the Supplementary Methods and Supplementary Table 1.

**Soil moisture correction and mask**. L-band radar backscatter is somewhat sensitive to soil moisture, which can enhance backscatter relative to dry conditions and reduce the distinction between wooded and bare areas[44]. The ALOS PALSAR mosaic product includes some images acquired outside the dry season (typically May- Nov) meaning that some wet season imagery is present in the data. The resultant seasonal soil moisture effects combined with the effect of the local hydrology of drainage lines and flood plains needs to be accounted for in multi-temporal AGC estimation. To reduce the effect of this on the estimates of biomass and biomass change, we undertook the following procedures: First, we developed a statistical model that applies a small correction to the estimated change in backscatter in areas where the estimated soil moisture was different between years. The model provides differential corrections according to the estimated AGC density, following the logic of the Water Cloud Model[56], with lower density areas more like to be susceptible to moisture effects (Supplementary Fig. 8 and Fig. 9). Thus, as a secondary precaution we do not include non-woodland lands (<10 MgC ha$^{-1}$ in 2007) in our change analysis as soil effects in these areas are likely to be particularly strong[8,44]. Third, areas where soil moisture varied markedly between 2007 and 2010 observations (e.g. when the data was collected at very different times of the year) were removed from the analysis (2% of land area). Finally, we exclude areas likely to have elevated soil moisture, or be seasonally flooded, including areas near rivers, water bodies, deltas and irrigated croplands (11% of the total land area) (Supplementary Fig. 7). A more detailed description of the soil moisture correction and masking procedure is located in the Supplementary Methods.

**Land-cover change definitions**. The multi-temporal AGC maps were to estimate the occurrence of the four land-cover-change (LCC) processes of interest, identified by comparing pixel carbon densities in 2007 and 2010. Areas where carbon stocks were found to be decreasing were classified according to whether they were symptomatic of deforestation (a reduction in wooded area), degradation (a reduction in biomass density), or whether losses were of a low intensity, symptomatic of minor disturbance. This distinction is designed to provide information on the manner and causes of change following Ryan et al. (2014).

Deforestation (more properly, the loss of wooded areas), is used to refer to a scenario where a pixel loses more than 20% of its biomass and moves from above the forest or woodland threshold in 2007, to below the threshold in 2010. The woodland- non woodland threshold is defined as 10 MgC ha$^{-1}$, based on data from the Tanzanian field plots where a biomass density of 10 MgC ha$^{-1}$ was equivalent to a tree canopy cover of 10–15%[38] meaning that our definition of woodland is similar to part of the United Nations Food and Agricultural Organisation's (FAO) forest definition[57]. These disturbances are likely to result from agricultural clearances—both small scale and commercial—or urban expansion[12]. Our definition of deforestation is designed to better capture areas converted to small scale agriculture by allowing for some residual woody biomass in the post clearance land cover - a common feature of the typically low-input, shifting agriculture that is widely practised in the region[13]. Shifting cultivators commonly leave large trees standing in their fields due to the disproportionate effort involved in their felling, or because the tree provides other ecosystem services, but as the resultant land cover is primarily agricultural, such land is properly classified as deforested.

Degradation is defined as reduction in carbon density to <80% of its 2007 value, whilst remaining above the woodland threshold. Previous studies[12,30] have shown that changes >20% of the 2007 AGC value are often caused by timber extraction and harvesting for charcoal production, as well as many other livelihood activities. Lower intensity losses where AGC is reduced by <20% are not considered as degradation or deforestation and instead are classified as minor losses, as they are often caused by quasi natural processes such as fire and tree mortality.

Gains in AGC stocks reflect the growth of woody vegetation and are limited to areas that were already wooded in 2007. This will include both the growth of mature, intact vegetation and areas re-growing following disturbance; hence, we sometimes refer to this process using the term (re)growth. We do not evaluate re- or afforestation as we exclude areas that are non-woodland in 2007.

**Probabilistic change detection**. To estimate the area and carbon emissions associated with each LCC process, we adopt a probabilistic approach, rather than a binary classification of change according to the most likely LCC type, e.g. degraded / not degraded. A probabilistic approach is ideally suited to biomass maps derived from synthetic aperture radar (SAR) imagery given the noise-like phenomenon of speckle that is inherent to this type of data, and which leads to high RMSE. Speckle arises because of interference between the signal from scatterers within a pixel, and leads to a well characterised distribution of observed backscatter, even over homogenous areas[53]. Any error distributions in backscatter will also be present in biomass maps, and could signal false LCC events when biomass maps from different time points are compared directly. If not accounted for, speckle would lead to an overestimation of land cover change; thus, we developed a statistical model that accounts for both the speckle-induced changes in backscatter, and the uncertainty on the regression model used to convert backscatter to biomass.

To account for speckle-induced changes in backscatter, and the synergy with errors in the regression between backscatter and biomass, we statistically model the expected effect of speckle through a simulation procedure. We first simulate 10,000 possible values (realisations) of backscatter ($\gamma^0$) following a gamma distribution; $\gamma^0 \sim \Gamma(k,\theta)$[53]. The parameters of the speckle model $k$ and $\theta$ were estimated empirically using backscatter data extracted from 20 pseudo homogeneous areas (area = 3–98 km$^2$; mean estimated AGC = 0–45 MgC ha$^{-1}$), chosen because the vegetation appeared homogenous, and their remoteness or inaccessibility suggested low levels of human disturbance. In each location we fitted gamma distributions to the observed backscatter distributions and estimated the parameters $k$ and $\theta$. As expected[53], $\theta$, the scale parameter, was a linear function of the mean backscatter, $\overline{\gamma^0}$, such that $\theta = \alpha\overline{\gamma^0} + \beta$, with $\alpha$ and $\beta$ estimated as 0.0134 ± 0.004 (mean ± SE) and 0.0001 ± 0.0001, based on a OLS regression fit ($R^2 = 0.49$, RMSE = 0.00011). The shape parameter, $k$, is effectively the equivalent number of looks in a PALSAR 25 m mosaic, and was calculated by dividing the observed, or expected backscatter value, by the scale parameter. As some of the variance in these areas will be due to vegetation heterogeneity, this approach is likely to overestimate the contribution of speckle to the observed variability in backscatter, resulting in conservative estimates of land cover change.

Each simulated value of $\gamma^0$ was then used to simulate 200 realisations of woody biomass, $B$, with $B \sim N(\mu, \sigma^2)$, where $\mu$ is given by the regression equation, and $\sigma^2$ is the standard error of the model. These two simulations give 2,000,000 possible values of AGC for each observation of backscatter ($\gamma^0$). The simulations are repeated for the full range of observed combinations of $\gamma^0$ in 2007 and 2010, and for each combination, the proportion of times that the simulated values met the land cover change criteria (Eqs. 1–6) is used as

the probability that the change has occurred:

$$P(\text{deforested}) = P(B_{07} \geq 10)P(B_{10}<10)P([B_{10}/B_{07}]<0.8) \tag{1}$$

$$P(\text{degraded}) = P(B_{07}>10)P(B_{10} \geq 10)P([B_{10}/B_{07}]<0.8) \tag{2}$$

$$P(\text{gain}) = P(B_{07} \geq 10)P(B_{10} \geq 10)P(B_{10}>B_{07}) \tag{3}$$

$$P(\text{minor losses}) = P(B_{07} \geq 10)P(B_{07}>B_{10})P([B_{10}/B_{07}] \geq 0.8) \tag{4}$$

There is also the probability that the pixel was wooded, or non-wooded in 2007:

$$P(\text{wooded}) = P(B_{07} \geq 10) \tag{5}$$

$$P(\text{non} - \text{wooded}) = P(B_{07}<10) \tag{6}$$

These simulations were used to create lookup tables of the probability for each land cover change given the observed backscatter in 2007 and 2010 (Fig. 7), which are then used to apply probabilities of each to whole study area. Our approach is therefore not to draw an arbitrary threshold between areas where we are confident a change has occurred and areas where we are not, but to instead represent each potential change event according to the probability that it is real and correctly classified.

In practise, the probability of a change having occurred increases with the size of the observed change in biomass (Fig. 7), unless the AGC in 2007 or 2010 is close to the 10 MgC ha$^{-1}$ threshold used to classify wooded/ non-wooded lands and separate deforestation from degradation. When the observed 2007 AGC is near the threshold, there is a high chance that a pixel was not wooded in the first place. For example, in the case of a single pixel that was estimated at 25 MgC ha$^{-1}$ in 2007, but only 16 MgC ha$^{-1}$ in 2010 (Fig. 7d–f)—a change that is consistent with our definition of degradation—we assign that pixel a probability of 0.75 [95% CI: 0.71–0.84; see next section for explanation how these CI values were derived] that it was in fact degraded. This is because we account for the probability that - due to speckle noise and uncertainties on the regression - the reduction was only a minor loss, and the even smaller probability that the loss is actually a gain. Similarly, if we consider a pixel in 2007 with the average AGC of 24 MgC ha$^{-1}$ which increased in biomass by 4 MgC ha$^{-1}$ over the 3 years then we assign the pixel a gain probability of 0.79 [0.75–0.85].

**Quantifying land cover and carbon stock changes**. The total area affected by each LCC is calculated by summing the probabilities of each LCC having occurred in a single pixel (Fig. 7), which is appropriate given there is no spatial pattern to speckle. To estimate the carbon stock changes caused by each of the LCC processes of interest, the observed per pixel changes in AGC stocks between 2007 and 2010 were multiplied by the probability that each LCC type has occurred to produce a weighted estimate of biomass change for each type in each pixel.

**Uncertainties**. Uncertainties on all quantities were estimated through the propagation of the uncertainty in the biomass-backscatter relationship, including the bias. We employed a 5000 × 2-fold cross-validation procedure[8], withholding half of the ground data used to calibrate the radar data, and using the remainder to generate the biomass-backscatter relationship. This uncertainty procedure was applied to a random subsample of 5% study area, comprising 2000 × 100 km$^2$ areas randomly distributed across the study area. For each area, we calculated all derived quantities, retaining the 2.5th and 97.5th percentiles of the 5000 estimates, and using these 95% CI to approximate the uncertainties over the whole study area (Supplementary Fig. 10). A more detailed explanation of our approach to quantifying changes and the associated uncertainties is located in the Supplementary Methods.

**Comparison to existing estimates**. We compare our wooded area estimates to the estimated forest area in 2005 from the FAO Global Forest Resources Assessment 2015 (FAO-FRA)[22]—which includes updated values for the previous assessments covering our study period—and the Hansen et al. (2013) dataset[4], which provides estimates of percentage tree cover for the year 2000 (https://earthenginepartners.appspot.com/science-2013-global-forest). In the Hansen dataset, trees are defined as "all vegetation taller than 5 m in height" and are presented as the percentage cover per 30 m grid cell (0–100%). As Hansen is a tree cover data set, and not a forest cover dataset, we initially defined wooded lands in the Hansen dataset as being any area with a tree canopy cover ≥1%. This threshold was chosen based on a visual analysis of the Hansen tree cover and forest loss dataset which revealed the presence of loss events (2007–2010) in areas with a tree canopy cover of <10% in 2000, indicating that these areas should be considered as wooded in all subsequent comparisons. To increase comparability, we include only those pixels that had not been cleared as of 2007. We also calculated woodland area using a 10% canopy

cover threshold to mimic the definition of used by FAO and the one adhered to in this study

Our carbon stock estimates were compared to those in the FAO-FRA dataset—which also includes information on carbon stock changes (Supplementary Table 3)—and to the satellite derived stock estimates from Baccini et al., (2012) and Saatchi et al., (2011), and the more recent dataset of Avitabile et al. (2016) which integrates these pan-tropical datasets into a 1 km resolution AGB map (carbon stocks assumed to be 0.47 of biomass) using an independent reference dataset of field observations and high-resolution biomass maps (Supplementary Fig. 2). The Baccini study reports AGC stocks for the period 2007–2008 and is cropped to the extent of the tropics (23.4N–23.4S), whereas the Saatchi estimates refer to the early 2000s; therefore we compare these stocks to our 2007 AGC map.

Our deforestation estimates were compared to the FAO-FRA forest area change statistics for 2005–2010, and to Hansen (technically tree cover loss). For the former, we calculated change *pro rata*, whereas for the latter, we extracted only those changes that occurred between 2007 and 2010 and divided these by the

estimated wooded area in 2007, which was any area with a tree canopy cover ≥1%. We repeated this using a 10% wooded area threshold as the denominator to examine how differences in tree cover influence the reported rates of change and the resultant comparisons.

In order to compare the location of detected deforestation to the Hansen dataset, we analysed only those pixels from this study where deforestation or degradation is the most likely outcome. All three datasets (Hansen, deforestation, degradation) were aggregated to 9 ha (300 m × 300 m) to ease the processing time given the large extent of the study region, but also to account for any minor geo-location differences between to two datasets. The absolute differences in the proportion of each 9 ha pixel that was deforested is shown for the entire study region in Supplementary Fig. 5, and is analysed by the baseline AGC density and the woody cover in 2007 according to Hansen et al. (2013) in Supplementary Fig. 6.

**Data availability**. Data are available from the University of Edinburgh DataShare service at the following address: https://datashare.is.ed.ac.uk/handle/10283/3059.

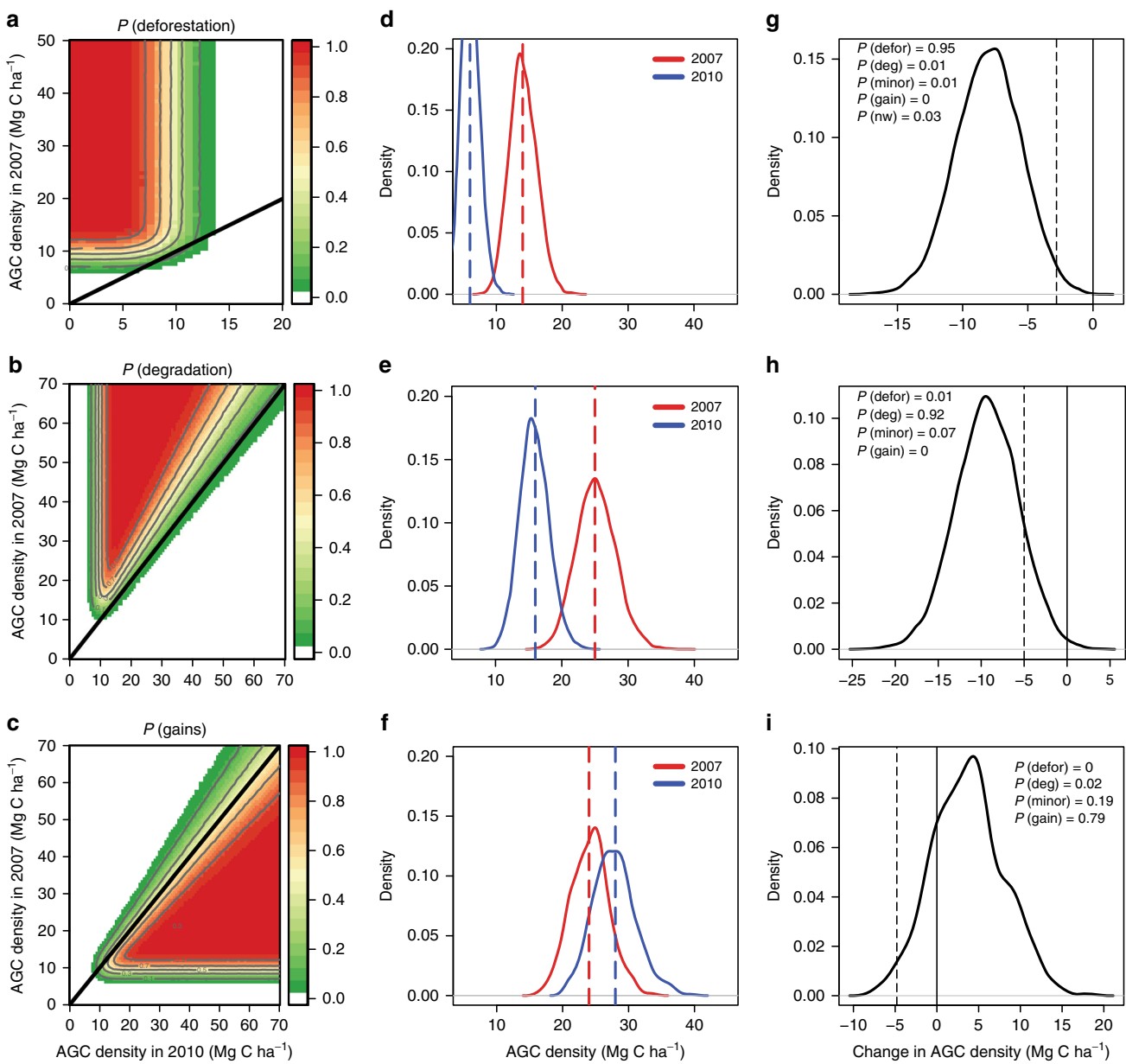

**Fig. 7** Approach to probabilistic change detection. **a–c** Matrices used to estimate the probability that (**a**) deforestation, (**b**) degradation or (**c**) a gain in biomass has occurred, given observed values of AGC in 2007 (*y*-axis) and 2010 (*x*-axis). Note the narrower scale on the axes of the deforestation in plot (**a**). The solid black line is the 1:1 line. The central (**d–f**) and right hand columns (**g–i**) provide illustrative examples of the probabilistic approach for three different change scenarios. This includes (**d–f**) the distribution of possible biomass values for 2007 and 2010 due to speckle and the regression model (observed values shown with dashed vertical lines) for three scenarios of change. The right hand panels show (**g–i**) the difference between the two distributions and the associated probabilities. The dashed vertical line indicates the <20% loss threshold used to separate deforestation and degradation from minor losses. The term P(nw) in Fig. 7g refers to the probability that the pixel was not wooded in 2007

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

## Acknowledgements

This work was partially funded by the Luc Hoffmann Institute at WWF-World Wide Fund for Nature (Project Number: 10002150). IMM's time was also supported by the Natural Environment Research Council (NERC) and Department for International Development (DfID) funded Understanding the Impacts of the Current El Niño programme (NE/P004725/1). CMR's time was supported by the NERC funded SEOSAW project (NE/P008755/1) and the ACES project, NE/K010395/1. ACES was funded with support from the Ecosystem Services for Poverty Alleviation (ESPA) programme. The ESPA programme is funded by the Department for International Development (DFID), the Economic and Social Research Council (ESRC) and the Natural Environment Research Council (NERC). We thank Mathew Williams and Aidan Keane at the University of Edinburgh for their advice and help. Rebecca Stedham and Yaqing Gou assisted with data processing. The original radar data were provided by JAXA, for which we are very grateful. This work is part of the 4th and 6th Research Announcement for the Advanced Land Observation Satellite-2 (ALOS-2). The field data from Malawi was provided by Steve Makungwa of Lilongwe University of Agriculture & Natural Resources, and was collected as part of the JICA funded Forest Preservation Survey. The field data from Tanzania was collected by MCDI, as part of a project led by Steve Ball, Jaspar Makala and Mathew Williams, and funded by the Royal Norwegian Embassy in Dar es Salam. Acknowledgements for the collection of the field data in Mozambique can be found in ref[8]. We are very grateful for the hard work of all those involved in the field data collection.

## Author contributions

I.M.M. and C.M.R. conceived the idea for the paper, and devised the methodology with input from E.M.; I.M.M. processed and analysed the data. C.M.R. developed the speckle model and probabilistic approach for detecting change with input from I.M.M.; I.M.M. and C.M.R. led the writing of the manuscript with input from E.M. All authors gave final approval for submission.

## Additional information

**Competing interests:** The authors declare no competing interests.

