## [Peer Review File · Nature Communications]

Reviewers' comments:

Reviewer #1 (Remarks to the Author):

This is a well-written paper, based on a rigorous methodology and that provides original insights on the role that woodlands may play globally in the carbon cycle. This paper has thus the potential to be of interest to a wide readership (this journal would be a proper outlet for this work). I have only few comments listed below.

From a methodological point of view, I much appreciated the efforts done to account for the uncertainties associated with the ALOS products. For instance, correcting for the speckle effects is an important step that has not been done systematically in the past. Also, I liked the way the authors have accounted for uncertainties in the degradation/deforestation classification. However, I still have some concerns on the methodology:

- (i) uncertainties from field estimates have not been accounted for, which may lead to strong uncertainties in the final product (especially given the small size of some plots).
- (ii) I did not understand to which Monte-Carlo (MC) procedure the authors referred in the section 2.3. (SI Line 263), I guess to the cross-validation approach which is not really a MC procedure ?!
- (iii) JAXA has recognized that a global decrease in intensity occurs in the ALOS product from 2007 to 2010. How this general decrease may have impacted the results found in this study ?

Another concern is the lack of discussion on the potential role of climate change in the long-term carbon dynamics of these woodlands. Do we have any idea of the expected climate trends over these areas and of their hypothesized effects on carbon dynamics ?

Is there any assumption on why the Avitabile et al. map underestimates/overestimates the Carbon stocks of woodlands in low/high biomass areas ? Can we expect a similar bias in other woodlands or in dense forests ?

Just my thought : I am rather impressed by the finding that the studied woodlands account for 4 - 10% of the current estimated gross tropical land-use emissions. It seems huge !

Line 213-214 : The way the areas of $AGC < 10 \text{ MgC/ha}$ were removed is unclear. If soil moisture mostly affects the backscatter in those areas (I partly agree), filtering out these areas based on their backscatter may be dangerous. I found the use of the ECV moisture index to be more convincing (by the way, please provide a reference or a weblink).

Line 209 : I personally do not think that the L-band backscatter is anymore really sensitive to carbon density above 75 MgC/ha (the above limit of 100 MgC/ha is a bit dangerous). Figure S2 only shows that no obvious saturation occurs up to 60 MgC/ha .

Lines 216-217 ; How exactly the difference in the timing was accounted for ?

Specific comments :

Line 14 : Indicate the area extent ?

Line 15-16 : It would be better to report these estimates (0.43 and -0.41) in PgC/yr

Line 17 : For information, a recently published paper (Pearson et al. 2017 Carbon Balance Manage 12 :3) also showed that degradation may release more C than deforestation (in 28 of 74 countries).

Line 56 : It would have been nice to see the location of field plots on the Fig. 1, or at least in the SI.

Line 60 : See also Mermoz et al. 2015, 2016 RSE

Line 74 : « in » duplicated

Lines 137-141 : Can we also hypothesize that a more favorable climate explain the gain in Biomass when human disturbances are negligible (as observed at the north of the Congo basin) ?

Reviewer #2 (Remarks to the Author):

Major comment

I have reviewed the paper untitled "Carbon losses from deforestation and widespread degradation offset by extensive growth in African woodlands" by McNicol et al. In their study, the authors have derived woodland aboveground carbon stock maps from 2007 to 2010 for seven countries in Southern Africa from radar satellite images (L-band backscatter from ALOS PALSAR). From the time-series they obtained, they provided estimates of deforestation, degradation and (re)growth for the time-period 2007-2010. They intended to show that (i) degradation was widespread and was the major source of carbon loss compared to deforestation, and (ii) large carbon loss was offset by large woodland (re)growth.

The topic is really interesting and scientifically timely. The objective is to find a method to monitor changes in forest carbon stocks using remote-sensing data, in order to help defining strategies to mitigate climate-change and conserve tropical forests and their associated biodiversity more efficiently. I want to underline the quality of the manuscript (text, maps and figures). The supplementary materials are also very well presented.

Nonetheless, I think that the study is too regional and too much focused on dry forests to be accepted in a journal such as Nature Communications. The L-band radar signal is known to saturate at high biomass values and can thus difficultly be used for moist tropical forests with higher biomass values. This prevent the approach to be used at continental or global scale.

I also have a major concern regarding the analysis that prevents the article to be accepted for publication. When looking at Figure S2 (which shows the most important step in the analysis), we can see that the relationship between aboveground carbon density (AGC) and radar backscatter is not strong enough to be able to measure small changes of AGC on a short period of time, such as the three years period between 2007 and 2010. On Fig. S2, we can see a large variability around the mean: for a given value of radar backscatter, there is a very large range of possible values for AGC. This is confirmed by the relatively low R2 (58%) and the high RMSE (8.73 MgC/ha) of the relationship. This RMSE value must be compared to the mean AGC which is of 24.6 MgC/ha (l. 75). This means that we have an uncertainty of about 35% around the AGC values, which is big.

On the contrary, the author estimated that the mean (re)growth is of about 1.30 Mg/ha/yr (l. 130). So for three years, the mean change would be of 3.90 Mg/ha/yr, which is 2 times lower than the RMSE.

As a consequence, it seems practically impossible to measure accurately small changes of AGC on such a short period of time between 2007-2010. To my point of view, it might also explain why the authors have estimated almost no change in the total AGC at the regional scales (6.1 PgC in 2007 and 6.2 PgC in 2010, l. 79). Gain and loss are in fact uncertainties and compensate at the regional scale.

One solution to overcome this problem would be to exclude the areas where the change is too small to be accurately measured. Only areas with large gain or loss (superior to the uncertainty) would be considered in the analysis. This approach is described for example in a recent study by Grinand et al. 2017 (see also Post et al., 2001) where the authors have estimated the soil organic carbon stock changes between two dates using remote-sensing data. This approach could be easily applied to your case-study as you have already computed the uncertainty with the Monte Carlo approach.

Minor comments

- At l. 16, define degradation in the summary (see definition l. 30).
- l. 49: specify the difference between growth (tree growth with carbon accumulation) and regrowth (forest regeneration).
- There is no discussion in the article about the possibility to extend the study to other regions (to other dry tropical regions and to moist tropical regions) or how to repeat the analysis in the future with new radar data (cf. L-band ALOS-2, L-band SAOCOM and P-band BIOMASS missions).

References

Grinand C., G. Le Maire, G. Vieilledent, H. Razakamanarivo, T. Razafimbelo, M. Bernoux. 2017. Estimating temporal changes in soil carbon stocks at the ecoregional scale in Madagascar using remote-sensing. *International Journal of Applied Earth Observation and Geoinformation*. 54: 1-14. [doi:10.1016/j.jag.2016.09.002].

Post, W. M., Izaurralde, R. C., Mann, L. K., & Bliss, N. (2001). Monitoring and verifying changes of organic carbon in soil. In *Storing Carbon in Agricultural Soils: A Multi-Purpose Environmental Strategy* (pp. 73-99). Springer Netherlands.

With the hope that this review will help the authors to improve their analysis and increase the impact of their research.

Best regards,

Reviewer #3 (Remarks to the Author):

I want to say from the outset that the work is substantial and if recast could yield an appropriately conservative use of PALSAR seasonal mosaics in estimating biomass change. However, the paper as presented overstates the dynamic due to a lack of good practice methods in estimating land change. Actual change in biomass is not validated. One can imagine letting such a method pass peer review and then we will have a multitude of difference-image studies reporting subtle changes, such as the gains presented here, without evidence of product accuracy from independent observations of the actual variable of interest – biomass change.

1) When validating land change, the accuracy of the change dynamic itself must be determined. The method presented does no such thing - a model uncertainty is presented, something that has nothing to do with accuracy. Repeat inventories over fixed sites that have experienced change are the only validation data I can imagine using to assess the true accuracy of a mapped biomass change product. Such data would preferably be probability-based in terms of locations within the study area. Again, this is not part of the study, only model uncertainty is presented.

2) The model is simplistic with high uncertainty. The method employs a filtered/modeled difference of a rather poorly calibrated ($\sim .5 R^2$) model to estimate changes in biomass. The combined uncertainties of this model over time surely lead to commission errors in estimating change. This limitation is not sufficiently discussed. For example, even a small bias one way or the other of one of the time periods will lead to huge changes regionally. We know that the errors of individual maps, whether land cover categories, percent woody cover, or biomass, result in false change and even multiply as errors in the individual maps are convolved. What you may be able to do is threshold a range of change that is clearly outside the combined uncertainty of the time-series models. This would allow for an argument that the changes are real, as the delta in biomass is

beyond the combined model uncertainties. Such a result, if it documented more deforestation than other products, would be a useful contribution. To apply such a model to every pixel without considering this basic limitation is to introduce and ignore considerable error.

3) The idea that a single backscatter feature can accurately estimate such a complicated biophysical variable is not realistic. We are to assume that soil moisture impacts have been accounted for, and I do appreciate the examples of wetlands introducing a clear bias between PALSAR images through time. However, this implies that the only areas with this problem are the wetlands. If that were true, a very accurate map of wetlands is needed, with its accuracy documented. However, that is not the main limitation, as the authors cannot account for the impacts of sources of backscatter outside of wetlands and not due directly to woody structure. What if one year was considerably wetter than another, or if the preceding week or day of image acquisition was wetter? You could easily get a regional-scale bias between El Nino and non-El Nino years, and even more variation due to short-term antecedent weather for any year. To assume that only wetlands are impacted by the soil moisture issue is incorrect.

4) What about PALSAR calibration and changes over time? Just curious, but that would be something to present if using a fixed model over multiple years. There is a reference to calibration, but it is critical to the use of turn-key models and should be elaborated upon.

In summation, continuous variables have sources of error – bias and precision – that when used in a time-series create a range of differences within which no estimation of change should be made. That is the first limitation. The second and larger one is the lack of an independent validation data set for the variable being mapped – not just static biomass, but biomass change.

Response to Reviewer #1

[01] This is a well-written paper, based on a rigorous methodology and that provides original insights on the role that woodlands may play globally in the carbon cycle. This paper has thus the potential to be of interest to a wide readership (this journal would be a proper outlet for this work). I have only few comments listed below.

[02] From a methodological point of view, I much appreciated the efforts done to account for the uncertainties associated with the ALOS products. For instance, correcting for the speckle effects is an important step that has not been done systematically in the past. Also, I liked the way the authors have accounted for uncertainties in the degradation/deforestation classification.

Thank you for these very positive comments.

[03] However, I still have some concerns on the methodology: (i) uncertainties from field estimates have not been accounted for, which may lead to strong uncertainties in the final product (especially given the small size of some plots).

This is a good point. The main source of uncertainty introduced by the field estimates will be due to the choice of allometric equation, which typically adds a systematic bias to the estimates of woody biomass on a set of plots (Williams *et al.*, 2008; Ryan, 2009; Hill *et al.*, 2013). To quantify this, we re-created the biomass-backscatter model using plot biomass estimates generated using the pantropical model by Chave *et al.*, (2005, 2014) and a regional allometric models developed by Mugasha *et al.*, (2013) in Tanzanian woodlands, and compared them to the biomass maps created with the original Ryan *et al.*, (2011) allometric model.

The use of these alternative allometric models resulted in regional estimates of AGC storage that were 11% and 18% higher when using Chave *et al.* (2005) and Mugasha *et al.* (2013) respectively, but 5% lower when using Chave *et al.* (2014), although the latter estimate is within our range of uncertainty. Hence, our use of the Ryan *et al.* (2011) model, which was developed close to the majority of the field plots used for calibration, led to fairly conservative estimates of AGC stocks than the best alternative regionally developed model. Lower stocks are likely to lead to an underestimate of losses upon land cover change so, our results on this front are likely to be conservative too. We include this analysis in the Supplementary Methods section of the SI.

Small plots do tend to produce slightly more uncertain estimates of biomass for a given area, particularly when there are a few large trees. However, the majority our individual field plots are relatively large, with a median and mean plot size of ~0.6 ha and a mode of 1 ha. Tree-level errors (e.g. due to the allometric model and measurement errors) tend to average out in these larger plots (Chave *et al.*, 2004). Smaller plots, i.e. those <0.25 ha, comprise only 11 of the 137 sites, meaning their influence on the regression is likely to be relatively small.

That being said, plot measurement errors will account for some unexplained variance in the biomass-backscatter relationship. We now better account for this by including the regression uncertainty when calculating the probability of a land cover change (e.g. deforestation) as having occurred (see response to comment 31). Our error estimation procedure and the reported CIs, based around the creation of 5000 different biomass-backscatter regressions using different subsets of the data, will now also account for the random errors introduced by the small plots.

[04] (ii) I did not understand to which Monte-Carlo (MC) procedure the authors referred in the section 2.3. (SI Line 263), I guess to the cross-validation approach which is not really a MC procedure?!

You are quite correct; we are referring to the cross-validation procedure described in section 2.3. As the method involves repeated random sampling, it does have some characteristics of a Monte Carlo method, however you are correct that this is not really the right description, therefore, we now refer to it simply as the ‘cross validation procedure’.

10 [05] (iii) JAXA has recognized that a global decrease in intensity occurs in the ALOS product from 2007 to 2010. How this general decrease may have impacted the results found in this study?

This is an important point that reviewer 3 also raised [see comment 36]. The issue has been studied in two papers (Shimada *et al.*, 2009, 2014) and we also contacted the ALOS PALSAR mission scientists to ask if they had evidence of a decrease in intensity of the ALOS sensor. Both these papers and the mission scientist’s response suggest that there has been no change in the sensor characteristics relevant to our study period (see Fig R1). Indeed, the first conclusion of Shimada *et al.* (2014) is:

“The PALSAR remained stable (within 0.065 dB) over its lifetime (from 2006 to 2010) so changes in HV γ_0 over time could be attributed to changes in the land cover.”

20 As figure 3 in Shimada *et al.*, (2014) shows (Fig. R1), the transmission power of the PALSAR instrument appears to be very stable over the full mission lifetime; a conclusion confirmed through discussions with the lead author of the aforementioned paper. There is a slight decrease in power at Cycles 24 – 25 and Cycle 32 (the end of the mission lifetime), however these do not coincide with the mean acquisition date of the radar imagery used to create the four annual mosaics used here.

Figure R1 – The transmission power of ALOS PALSAR over the mission lifetime.

We have added the following text to the Methods section in the main text to make this clear:

“We assume that the biomass-backscatter relationship is consistent over time. The assumption of an invariant physical relationship between the remotely sensed observation and land surface is common among change detection studies using radar and other sensors (Mitchard et al., 2011; Ryan et al., 2012; Liu et al., 2015; Brandt et al., 2017). In our case, there is no reason to dispute this assumption given that soil moisture impacts have already been accounted for, and the stability of the sensor response has been verified (Shimada et al., 2014), meaning any changes in backscatter are more likely to be related to changes in land cover.”

10 [06] Another concern is the lack of discussion on the potential role of climate change in the long-term carbon dynamics of these woodlands. Do we have any idea of the expected climate trends over these areas and of their hypothesized effects on carbon dynamics?

We have reviewed the evidence for a climate and pCO₂ driven shift in the vegetation of the region in Ryan et al., (2016, Phil Trans Royal Soc). Although such shifts are predicted in several vegetation models, we concluded that because the climate trends and pCO₂ effects produce opposite effects it is currently beyond the state of the science to suggest whether tree growth will be enhanced or reduced by global change. We have added the following text at the end of the paper to make this point:

20 *“Continued monitoring of these systems is needed to evaluate the permanence of these land cover changes, and to evaluate the impacts of changes in climate and atmospheric CO₂ concentrations – drivers which are likely to have contrasting effects on woody cover over the next century”.*

[07] Is there any assumption on why the Avitabile et al. map underestimates/overestimates the carbon stocks of woodlands in low/high biomass areas? Can we expect a similar bias in other woodlands or in dense forests?

30 Ultimately the Avitabile et al. map is a fusion of the Saatchi et al. (2011) and Baccini et al. (2012) maps, locally anchored by field data. It therefore suffers from the same errors and biases as those underlying maps (Hill et al., 2013; Mitchard et al., 2013). These errors appear to be particularly large in our study region where there is very little field data for calibration. Thus, in central Mozambique, where it is near field plots that we contributed to the map, the Avitabile map appears reasonably unbiased, however these errors tend to become more apparent the further it is from field plots. It is unclear what drives this spatial variability, particularly the underestimation in lower biomass areas, however in most areas, the difference tends to be quite small with district level stocks typically within the margin of error.

40 The Baccini and Saatchi AGC maps are based on comparing stand height parameters retrieved by spaceborne LiDAR to field estimated stocks, the majority of which comes from ‘intact’ moist forests. This will lead to a systematic bias in areas where tree allometry is markedly different to the sites used for calibration. For example, tree height in savanna woodlands may be smaller for a given diameter than in tropical forests as they tend to allocate more resources to root biomass (Nzunda et al., 2014; McNicol et al., 2015). This is to allow rapid regrowth following frequent disturbance, which in savannas is more important than attaining a greater height than in denser ‘forest’ ecosystems. This may explain why Avitabile underestimates tree biomass in lower biomass areas, although it is purely speculation.

[08] Just my thought: I am rather impressed by the finding that the studied woodlands account for 4 - 10% of the current estimated gross tropical land-use emissions. It seems huge!

Yes this was a surprising result. Of course, there are errors on these global values that are hard to characterise.

[09] Line 213-214: The way the areas of $AGC < 10 \text{ MgC ha}^{-1}$ were removed is unclear. If soil moisture mostly affects the backscatter in those areas (I partly agree), filtering out these areas based on their backscatter may be dangerous. I found the use of the ECV moisture index to be more convincing

10 The reviewer raises a good point about the circularity with creating the mask. To simplify things, we have removed this mask completely, so the analysis now includes all values of AGC. The only mask used now is the one detailed in SI § 2.8 which includes waterbodies, flooded areas, and other areas where the biomass-backscatter relationship does not apply.

The removal of the 10 MgC ha^{-1} AGC mask makes no difference to the deforestation, degradation and gain estimates as the definition of these changes only considers areas $> 10 \text{ MgC ha}^{-1}$. It does change our estimates of the regional carbon stocks slightly, as the area $< 10 \text{ MgC ha}^{-1}$ (which we refer to as “other wooded land”; OWL) now makes a small contribution to the total. This is shown in Supplementary Table 4, with new columns for “biomass in “OWL2007” through to “biomass in OWL2010”. We acknowledge in the main text at Line

20 28 that we are less confident of the stocks and changes in these OWL areas (Tanase *et al.*, 2014).

[10] Line 209: I personally do not think that the L-band backscatter is anymore really sensitive to carbon density above 75 MgC ha^{-1} (the above limit of 100 MgC ha^{-1} is a bit dangerous). Figure S2 only shows that no obvious saturation occurs up to 60 Mg/C .

Yes, point taken. The saturation point is poorly known (Mitchard *et al.*, 2009; Lucas *et al.*, 2010) and that our dataset cannot diagnose the saturation point for our region as there are few plots with sufficiently high biomass. To assess the impact of the choice of saturation point, we conducted a sensitivity analysis of the regional biomass stocks and change estimates, varying the assumption of the saturation point (“AGC cap”) from 60 to 100 MgC ha^{-1} .

30

Figure R2 a) Aboveground carbon stocks in 2007 and b) their changes from 2007-2010 for the study region, using different assumption of the saturation of the biomass-backscatter relationship. The different shades indicate different assumptions of the carbon density (tC/ha) at which the relationship saturates. In the left hand panel, please note that the scale does not begin at zero.

This analysis showed that in 2007 only 0.15% of wooded pixels exceeded 60 MgC ha⁻¹, and only 0.004% exceeded the suggested saturation point of 75 MgC ha⁻¹. As such, progressively lowering the cap from the original 100 MgC ha⁻¹ limit resulted in a negligible decrease in AGC stocks and changes. Following the reviewer’s advice, we now implement a cap at 75 MgC ha⁻¹ after Mitchard *et al.* (2009), whereby any pixel >75 MgC ha⁻¹ is assumed to have a biomass of 75 MgC ha⁻¹.

We have edited the relevant text in the Methods section of the main text, and in the supplementary material have added the following:

“Any pixel with an AGC density >75 MgC ha⁻¹ was capped at this value in order to account for the decrease in sensitivity of L-band radar at higher AGC values leading to an eventual saturation in the backscatter response (Mitchard *et al.*, 2009). Varying the cap from 60 MgC ha⁻¹ to 100 MgC ha⁻¹ had a negligible effect on AGC stocks and changes owing to the relatively small number of these high biomass pixels”.

[11] Lines 216-217; How exactly the difference in the timing was accounted for?

Original text: “...and areas where differences in the timing of the radar acquisitions meant the soil moisture was very different between 2007 and 2010 observations”

The acquisition date of the radar imagery used to create the mosaics is provided with the mosaic dataset. We used this information to extract the estimated soil moisture from the ECV soil moisture time series. We have included a more detailed explanation of how the soil moisture maps were created in was created in SI § 2.8. We have modified the methods in the main text to simply say:

“However, areas where soil moisture varied markedly between 2007 and 2010 observations (e.g. when the data was collected at very different times of the year), were also removed from the analysis (2% of land area).”

[12] Lines 137-141: Can we also hypothesize that a more favourable climate explain the gain in Biomass when human disturbances are negligible (as observed at the north of the Congo basin)?

10 This might well be true; certainly there is evidence that growth rates in the wetter part of the region are almost double the driest parts (Frost, 1996). We have now included this as potential reason for some of the gains observed in the cited section of text. Also see text added as part of the response to comment [7] above.

Minor changes suggested by Reviewer #1

[13] I found the use of the ECV moisture index to be more convincing (by the way, please provide a reference or a weblink).

We have now included a weblink and citation for the ECV soil moisture product at first mention in section 2.8 of the SI

[14] Line 14: Indicate the area extent?

The area extent of wooded lands is now included in the summary

20 [15] Line 15⁻¹⁶ : It would be better to report these estimates (0.43 and -0.41) in PgC/yr
Altered to PgC yr⁻¹

[16] Line 17: For information, a recently published paper (Pearson et al. 2017 Carbon Balance Manage 12 :3) also showed that degradation may release more C than deforestation (in 28 of 74 countries).

We have added this reference, thanks.

[17] Line 56 : It would have been nice to see the location of field plots on the Fig. 1, or at least in the SI.

The location of the field plots have been added to the Fig. S1 in the SI

[18] Line 60 : See also Mermoz et al. 2015, 2016 RSE

We have added these references. Thanks.

30 [19] Line 74 : « in » duplicated

Removed.

Response to Reviewer #2

[20] ... The topic is really interesting and scientifically timely. The objective is to find a method to monitor changes in forest carbon stocks using remote-sensing data, in order to help defining strategies to mitigate climate-change and conserve tropical forests and their

associated biodiversity more efficiently. I want to underline the quality of the manuscript (text, maps and figures). The supplementary materials are also very well presented.

We thank the reviewer for their positive comments.

[21] Nonetheless, I think that the study is too regional and too much focused on dry forests to be accepted in a journal such as Nature Communications. The L-band radar signal is known to saturate at high biomass values and can thus difficultly be used for moist tropical forests with higher biomass values. This prevents the approach to be used at continental or global scale.

We argue that the paper has global relevance for two main reasons:

10 1) This method is suitable for use in all savannas and woodlands, as well as some dry forests. These dry tropical ecosystems are important in their own right as part of the global carbon cycle. This is true both in terms of the functioning of the intact vegetation, which is the main driver of interannual variability in the atmospheric CO₂ concentration (Poulter *et al.*, 2014, *Nature*), and for their role in land use emissions: we show in the paper that the gross emissions from land cover change in African woodlands are not too dissimilar to those from land cover change in the more spatially extensive Brazilian Amazon, and are equivalent to 4 - 10% of the current estimated gross tropical land-use emissions.

20 2) As discussed above, it is clearly true that L-band radar saturates at high biomass. As the reviewer points out, this means that the data source used here (ALOS PALSAR) cannot be used in high biomass tropical forests. However, our method is not restricted to L-band or ALOS data, and would work equally well for data from longer wavelength radars. As the reviewer points out below, the European Space Agency will launch a P-band radar around 2020 (BIOMASS), that is designed to estimate biomass in the wet tropics, and along with the planned L-band missions by NASA/ISRO and Argentina, there will soon be the potential for a global biomass observing system. The methods used here will be able to utilise such data to make global estimates of deforestation, degradation and regrowth.

30 We have included the Poulter et al. (2014) reference at Line 50, and have added in a section of text at the end of the manuscript about the possibility to extend the study to other dry and wet tropical regions using new radar data (e.g. L-band ALOS-2, L-band SAOCOM and P-band BIOMASS missions) in response to a similar comment from Reviewer 3 [28]:

“The methods presented here are not specific to the radar satellite used (ALOS PALSAR), and are applicable to longer wavelength radar sensors, including the P-band BIOMASS mission (Le Toan et al., 2011) which is designed to estimate biomass in more carbon dense moist tropical forests, and the planned L-band SAOCOM-1 and NISAR missions (Reiche et al., 2016)”

40 [22] I also have a major concern regarding the analysis that prevents the article to be accepted for publication. When looking at Figure S2 (which shows the most important step in the analysis), we can see that the relationship between aboveground carbon density (AGC) and radar backscatter is not strong enough to be able to measure small changes of AGC on a short period of time, such as the three years period between 2007 and 2010. On Fig. S2, we can see a large variability around the mean: for a given value of radar backscatter, there is a very large range of possible values for AGC. This is confirmed by the relatively low R² (58%) and the high RMSE (8.73 MgC ha⁻¹) of the relationship. This RMSE value must be compared to the mean AGC which is of 24.6 MgC ha⁻¹ (l. 75). This means that we have an uncertainty of about 35% around the AGC values, which is big.

On the contrary, the author estimated that the mean (re)growth is of about 1.30 Mg/ha/yr (1.130). So for three years, the mean change would be of 3.90 Mg/ha/yr, which is 2 times lower than the RMSE.

The reviewer raises an important point here, but we think we have miscommunicated our methods, as we do account for the errors in the biomass-backscatter relationship in our uncertainty calculations, and *over large numbers of pixels* our approach is able to detect small changes, because random errors (i.e. the RMSE) cancel out. There are two key elements to the method the evaluation of the bias in the regression, and the use of a probabilistic approach to change detection:

10 Firstly, when looking at the relationship between biomass and backscatter, we are mostly interested in the bias as this is what affects large areas, whereas the RMSE cancels out over multiple pixels. A summary of the issue follows, but this point is explained more fully in (Ryan *et al.*, 2012).

In our case, the biomass-backscatter relationship has a RMSE of $\sim 8 \text{ MgC ha}^{-1}$, and a bias of $\sim 1 \text{ MgC ha}^{-1}$. As the reviewer points out, this means that small changes are difficult to detect for an individual pixel, as the RMSE is higher than many of the likely changes. However, the RMSE is a random error, mostly caused by speckle noise, and as such, it can be reduced by averaging over many pixels.

20 In our study, the smallest area we have reported results for is $\sim 400 \text{ km}^2$, which contains 640,000 pixels (n) (SI §2.7). If we assume the errors in each year are uncorrelated, then the total RMSE for the area is given as $\sqrt{\text{RMSE}^2 * n}$, which is 6800. If this area had the mean carbon density (23.8 MgC ha^{-1}) this error is 0.04% of the carbon stock. What this shows is that the RMSE becomes trivial for progressively larger areas, even though it is important for a single pixel.

30 There is of course a risk that the regression we have fitted is biased, that is, that the line is too high or too low, or the slope is wrong. This is captured in our bias statistic, which evaluates the absolute difference between the prediction of the model and the hold-out data points. The bias does not diminish with increasing number of pixels, and is the key measure of the ability of this method to accurately estimate biomass over a large area. In our case, it is around 1 MgC ha^{-1} , or 4% of the mean.

However, in this study, we carefully propagate the errors associated with the biomass-backscatter relationship, including the bias, by simulating of different regressions through to the biomass maps and estimates of change. In practice this leads to 95% CIs for biomass that range from 89% to 111% of the sum of the biomass for a 400 km^2 area.

40 *[23] One solution to overcome this problem would be to exclude the areas where the change is too small to be accurately measured. Only areas with large gain or loss (superior to the uncertainty) would be considered in the analysis. This approach is described for example in a recent study by Grinand et al. 2017 (see also Post et al., 2001) where the authors have estimated the soil organic carbon stock changes between two dates using remote-sensing data. This approach could be easily applied to your case-study as you have already computed the uncertainty with the Monte Carlo approach.*

We adopt a probabilistic approach to the estimation of deforestation, degradation and regrowth of individual pixels which is specifically designed to account for the random errors

that lead to the high RMSE. We first construct a probability distribution for the biomass of the pixel in 2007 and another distribution for the biomass in 2010 (Figure R3). This is done using the model of speckle noise presented in SI § 2.5, and now includes the uncertainties in the regression as well. As some of the variance in backscatter in the areas used to create the speckle model will be due to vegetation heterogeneity, our approach likely overestimates the contribution of speckle to the observed variability, resulting in conservative estimates of land cover change. Fig S3 in the supplementary material illustrates how the probability of a land cover change having occurred increases with the size of the observed change in biomass, (unless the AGC in 2007 and/or 2010 is close to the forest threshold - then there is uncertainty over whether a pixel is degraded or deforested (Fig. R3 bottom panels)).

Figure R3 (Now Figure S4 in SI) – An example of the probabilistic approach used to estimate the probability that deforestation, degradation or a gain in biomass has occurred in a single pixel. The left hand panels show the distribution of possible biomass values for 2007 and 2010 due to speckle and the regression model, with the right hand panels showing the difference between the two distributions. The top panels highlight a scenario consistent with degradation, the middle panels a slight gain, and the bottom panels a change consistent with deforestation. The dashed horizontal line indicates the <20% loss threshold used to separate deforestation and degradation from minor losses.

20

An example follows to illustrate the method. In the case of a single pixel that was 25 MgC ha⁻¹ in 2007, but only 16 MgC ha⁻¹ in 2010 (Fig. R3 top panels), we estimate a probability of 0.75 [95% CI: 0.71 – 0.84] that the area was degraded. This is because we account for the

smaller probability (due to speckle noise and uncertainties on the regression) that the reduction was actually <20% and only a minor loss, but also at the even smaller probability that the loss was actually a gain.

Similarly, if we consider the example of a pixel in 2007 with the regional average AGC of 24 MgC ha⁻¹ which increased in biomass by the regional average of 4 MgC ha⁻¹ over the 3 years (Fig S3, central panels), then we assign the pixel a gain probability of 0.69 [95% CI: 0.65 – 0.75].

10 Thus, we are essentially doing as the reviewer suggests, except that instead of classifying a change as “too small to be accurately measured”, we assign the pixel a probability that the change has occurred and use this probability value to weight the observed changes in biomass and area of change. This leaves the user of these maps free to select a threshold of probability most suitable for their purpose.

We understand that this is not the standard method, however we believe this is more appropriate given the speckle (noise) associated with this type of radar imagery.

We hope our explanation addresses the reviewer’s concerns. We have tried to make things clearer, particularly in SI §2.5 where we include a more detailed explanation of our approach using the above figure and excerpts from the text above.

20 [24] As a consequence, it seems practically impossible to measure accurately small changes of AGC on such a short period of time between 2007-2010. To my point of view, it might also explain why the authors have estimated almost no change in the total AGC at the regional scales (6.1 PgC in 2007 and 6.2 PgC in 2010, l. 79). Gain and loss are in fact uncertainties and compensate at the regional scale.

The issue of how we account for small changes has been discussed above.

There is clear evidence of strong spatial patterns in biomass change across the region, and these cannot have arisen from a random process. This spatial pattern mirrors the observations of previous studies (Mitchard & Flintrop, 2013; Liu *et al.*, 2015; Brandt *et al.*, 2017).

[25] At l. 16, define degradation in the summary (see definition l. 30).

We define it at first use in the main text due to tight restrictions on word count in the summary.

30 [26] l. 49: specify the difference between growth (tree growth with carbon accumulation) and regrowth (forest regeneration).

We now clearly define each of our land cover change scenarios in the methods section of the main text and make it clear that our analysis of gains will include both processes;

“4) Growth, where a pixel increases in biomass from 2007 to 2010. We use the expression (re)growth, as this process includes both woodland regrowing after disturbance, and biomass increases in intact woodland”

40 [27] There is no discussion in the article about the possibility to extend the study to other regions (to other dry tropical regions and to moist tropical regions) or how to repeat the analysis in the future with new radar data (cf. L-band ALOS-2, L-band SAOCOM and P-band BIOMASS missions).

We have added text to address this: see text added as part of the response to comment [22].

Response to reviewer #3

[28] I want to say from the outset that the work is substantial and if recast could yield an appropriately conservative use of PALSAR seasonal mosaics in estimating biomass change.

Thanks!

[29] However, the paper as presented overstates the dynamic due to a lack of good practice methods in estimating land change. Actual change in biomass is not validated. One can imagine letting such a method pass peer review and then we will have a multitude of difference-image studies reporting subtle changes, such as the gains presented here, without evidence of product accuracy from independent observations of the actual variable of interest – biomass change.

1) When validating land change, the accuracy of the change dynamic itself must be determined. The method presented does no such thing - a model uncertainty is presented, something that has nothing to do with accuracy. Repeat inventories over fixed sites that have experienced change are the only validation data I can imagine using to assess the true accuracy of a mapped biomass change product. Such data would preferably be probability-based in terms of locations within the study area. Again, this is not part of the study, only model uncertainty is presented.

We agree with the reviewer that corroboration of the estimates presented here with repeat *in situ* measurements over randomly located sites would be the perfect way to fully validate such an analysis. However, what the reviewer asks for is simply not feasible as there are no contemporaneous *in situ* or other independent measurements of biomass change that are representative of the study area.

Nevertheless, we would argue that it is not crucial to have independent estimates of the variable of interest recorded through time to generate remotely sensed estimates of change. Instead, you can use information on the relationship between the remotely sensed quantity and the variable of interest at one time point, and use physical theory to evaluate whether this is likely to hold over time. We argue that it is justified, and common practice, to make such an assumption of temporal constancy in the physical relationship between the land surface and the remotely sensed observation. Examples of publications which do so to map woody change include:

- Liu YY, et al. (2015) Recent reversal in loss of global terrestrial biomass. *Nature Climate Change*, 5, 470–474.
- Brandt M, et al. (2017) Human population growth offsets climate-driven increase in woody vegetation in sub-Saharan Africa. *Nature Ecology & Evolution*, 1, 81
- Marle MJE van, et al. (2016) Annual South American forest loss estimates based on passive microwave remote sensing (1990–2010). *Biogeosciences*, 13, 609–624

And for NPP changes:

- Nemani et al. (2003) Climate-Driven Increases in Global Terrestrial Net Primary Production from 1982 to 1999. *Science*, 300

In our case, the physical theory governing the biomass-radar backscatter relationship provides no reason to think that the relationship changes over time, once the impact of soil moisture has been accounted for (see SI § 2.8), and with appropriate checks on the stability of the sensor (see response to comment [6]). We've used this assumption as the basis for previously published work in a range of journals over the last six years, e.g.:

- Mitchard ETA, Saatchi SS, et al. (2011) Measuring biomass changes due to woody encroachment and deforestation/degradation in a forest–savanna boundary region of central Africa using multi-temporal L-band radar backscatter. *Remote Sensing of Environment*, 115, 2861–2873.
- Ryan CM, et al. (2012) Quantifying small-scale deforestation and forest degradation in African woodlands using radar imagery. *Global Change Biology*, 18, 243–257.
- Ryan CM, et al. (2014) Quantifying the causes of deforestation and degradation and creating transparent REDD+ baselines. *Applied Geography*, 53, 45–54.
- 10 • Joshi N, Mitchard ET, et al. (2015) Mapping dynamics of deforestation and forest degradation in tropical forests using radar satellite data. *Environmental Research Letters*, 10, 34014.
- Collins MB, Mitchard ETA (2015) Integrated radar and lidar analysis reveals extensive loss of remaining intact forest on Sumatra 2007–2010. *Biogeosciences*, 12, 6637–6653.

The only difference here is the larger scale of the analysis.

Of course, remotely sensed estimates of change need to be corroborated by *in situ* estimates, and other methods before they can be treated as definitive, as we stated at the outset of our response. We assert that the conclusions suggested by our results should be the motivation to
 20 collect the kind of *in situ* data the reviewer asks for, given the current absence of such data. We have added some text to that effect at Line 208:

“Future monitoring efforts will ideally incorporate repeat in situ observations of biomass growth and loss to corroborate remotely sensed estimates of change – something that was not possible here due to the lack of any contemporaneous large scale measurements of biomass change.

We have also added an additional bit of text to the main methods to make our assumptions clearer:

30 *“We assume that the biomass-backscatter relationship is consistent over time. The assumption of an invariant physical relationship between the remotely sensed observation and land surface is common among change detection studies using radar and other sensors (Mitchard et al., 2011; Ryan et al., 2012; Liu et al., 2015; Brandt et al., 2017). In our case, there is no reason to dispute this assumption given that soil moisture impacts have already been accounted for, and the stability of the sensor response has been verified (Shimada et al., 2014), meaning any changes in backscatter are more likely to be related to changes in land cover.”*

[30] 2) The model is simplistic with high uncertainty. The method employs a filtered/modeled difference of a rather poorly calibrated (~.5 R2) model to estimate changes in biomass. The combined uncertainties of this model over time surely lead to commission errors in estimating change. This limitation is not sufficiently discussed. For example, even
 40 a small bias one way or the other of one of the time periods will lead to huge changes regionally. We know that the errors of individual maps, whether land cover categories, percent woody cover, or biomass, result in false change and even multiply as errors in the individual maps are convolved. What you may be able to do is threshold a range of change that is clearly outside the combined uncertainty of the time-series models. This would allow for an argument that the changes are real, as the delta in biomass is beyond the combined model uncertainties. Such a result, if it documented more deforestation than other products,

would be a useful contribution. To apply such a model to every pixel without considering this basic limitation is to introduce and ignore considerable error.

We'd like to thank the reviewer for this comment which made us rethink our approach to incorporating the uncertainty in the biomass-backscatter relationship into the uncertainty of the estimates of biomass change. The resulted in a new approach and alteration to our method, whereby the regression error is now combined with the speckle distribution when assessing the probability of a land cover change (e.g. deforestation/degradation/regrowth) having occurred.

10 Errors of commission, or false positives, are a particular hazard of using radar remote sensing data, which includes a large amount of quasi-random fluctuations generated by the noise-like phenomenon referred to as speckle (Oliver & Quegan, 1998). The statistical properties of speckle are, however, well understood, and as noted by Reviewer 1 and as explained in our response to Reviewer 2 [23 & 24], we take considerable effort to model this effect. Thanks to Reviewer 3's comment no. 31, we now combine these speckle-induced uncertainties with the uncertainties in the biomass-backscatter relationship.

20 Our approach was not to draw an arbitrary threshold between areas where we are confident a change has occurred, and areas where we are not, but to instead estimate the probability that change has occurred given the data and the uncertainties. We use this probability to weight the observed changes in biomass and area of change. As such, we take great care to avoid false positives by weighting all but the largest changes in biomass. We believe this is more appropriate approach that the threshold the reviewer suggests given the speckle (noise) associated with SAR observations.

This is now more fully described in section 2.6 of the SI (see response to similar comments [23 +24]), and a new section in the main paper methods:

30 *"To estimate the area and carbon emissions associated with each LCC process, we adopt a probabilistic approach that is specifically designed to account for the random errors that contribute to the high RMSE, rather than a binary classification of change according to the most likely LCC scenario, e.g. degraded / not degraded. A probabilistic approach is suited to biomass maps derived from synthetic aperture radar (SAR) imagery given the noise-like phenomenon of speckle that is inherent to this type of data. Speckle arises because of interference between the signal from scatterers within a pixel, and leads to a well characterised distribution of observed backscatter, even over homogenous areas(Oliver & Quegan, 1998). Any error distributions in backscatter will also present in biomass maps, and could signal false LCC events when biomass maps from different time points are compared directly. In not accounted for, speckle would lead to an overestimation of land cover change; thus, we developed a statistical model that accounts for both the speckle-induced changes in backscatter, and the uncertainty on the regression model used to convert backscatter to biomass (see Supplementary Information section 2.6). This model conservatively estimates the probability that a real LCC has occurred"*

40 We would also point out that in the current work we use more ground data than ever before and carefully propagate the errors associated with the biomass-backscatter relationship through to the biomass maps and estimates of change. We also note that the relationship is about as good as it gets for calibration of biomass-backscatter curves, which vary from 0.27 to 0.86 (Mitchard *et al.*, 2009, 2011; Ryan *et al.*, 2012; Carreiras *et al.*, 2013; Mermoz *et al.*, 2014), which mostly include studies at single sites, and with flat terrain. In contrast our

calibration involves three sites and very steep topography. We also use natural units and a linear regression whereas all of the studies cited above use dB (i.e. log transformations) and fit a more complicated model with more parameters.

[31] 3) The idea that a single backscatter feature can accurately estimate such a complicated biophysical variable is not realistic.

We agree, but in essence we think backscatter is a good proxy for biomass within the study area considered here (Mitchard *et al.*, 2009; Ryan *et al.*, 2012; Tanase *et al.*, 2014). The consistency of the biomass-backscatter relationship across three sites (Fig S2) supports this. Underlying this issue is the fact that backscatter is not a ‘direct’ measure of biomass
10 (Woodhouse *et al.*, 2012) but is instead related to the number of scatters of a characteristic size. This is in turn related to the number of branches and small stems in the ground area under study. The relationship between the number of stems and branches appears to be well correlated with biomass across our three sites, and beyond (Mermoz *et al.*, 2014; Tanase *et al.*, 2014), but there may well be areas where this changes from the mean. However, because the key results hinge on looking for differences over time, we believe that many of these errors will cancel out.

[32] We are to assume that soil moisture impacts have been accounted for, and I do appreciate the examples of wetlands introducing a clear bias between PALSAR images through time. However, this implies that the only areas with this problem are the wetlands. If
20 that were true, a very accurate map of wetlands is needed, with its accuracy documented.

As such a map does not exist, we have taken considerable care and effort to mask out areas that *might* be wetlands. This means we have excluded from our study all areas within 1-3 km of a lake or river (depending on the size) to account for this potential issue.

[33] ... However, that is not the main limitation, as the authors cannot account for the impacts of sources of backscatter outside of wetlands and not due directly to woody structure. What if one year was considerably wetter than another, or if the preceding week or day of image acquisition was wetter? You could easily get a regional-scale bias between El Nino and non-El Nino years, and even more variation due to short-term antecedent weather for any year. To assume that only wetlands are impacted by the soil moisture issue is incorrect.

30 The reviewer is clearly correct that soil moisture can influence the backscatter signal. We take several steps to avoid this influencing the results.

Firstly, we mask out areas which are likely to have fluctuations in soil moisture – areas near rivers and lakes, and those classified in the land cover map as irrigated or flooded.

Secondly, we also mask out areas where the soil moisture is markedly different in 2007 and 2010 (Section 2.8). This was done specifically to account for areas where the radar data was acquired at very different times of the year, or when the preceding week or day was considerably wetter, resulting in clear moisture induced ‘striping’ in the data (*sensu* Lucas *et al.* 2010). As such we do not assume that it is only wetland areas that are impacted by soil moisture.

40 Thirdly, the vast majority of the imagery used in the mosaic comes from the mid to late dry season where the observations are relatively insensitive to difference in precipitation in the preceding rainy season, as the ground is very dry by then. According to Ryan *et al.*, (2017), the end of the wet season in our study region tends to occur around day of year 150 and the next rainy season begins from DOY 270. Most of our data comes from the dry period between these dates. For 2007, mean DOY of acquisition date is 213 (172 – 253; range equals

±2 SD), avoiding the wet periods, and the same is true for 2010 (mean acquisition date 227 (177 - 278)).

[34] 4) What about PALSAR calibration and changes over time? Just curious, but that would be something to present if using a fixed model over multiple years. There is a reference to calibration, but it is critical to the use of turn-key models and should be elaborated upon.

A very good point to raise; as noted at the end of our response to one of your earlier comments [30] and in response to reviewer 1 [6], we have included new evidence for the temporal constancy of the PALSAR observations into the main text.

10 “..there is no reason to dispute this supposition [assumption of temporal constancy in the biomass-backscatter relationship] given that soil moisture impacts have largely been accounted for, and the stability of the sensor response over time has been verified (Shimada et al., 2014), meaning that any changes in backscatter are more likely to be related to changes in land cover.”

[35] In summation, continuous variables have sources of error – bias and precision – that when used in a time-series create a range of differences within which no estimation of change should be made. That is the first limitation.

To summarise our response here, we agree with the principle, but our approach to dealing with this differs to that proposed by the reviewer: rather than declare an arbitrary
20 classification of “no change detection possible”, we calculate the probability of each land cover change having occurred and use these to weight the changes in biomass.

[36] ... The second and larger one is the lack of an independent validation data set for the variable being mapped – not just static biomass, but biomass change.

We believe such a validation data set is not needed for the reasons set out above. Our assumption that the biomass-backscatter relationship holds over time is now more clearly discussed in the methods of the main paper and in SI § 2.5, and we have taken steps to account for possible changes in soil moisture, and provided evidence that the sensor response has not changed over the study period. If the reviewer can suggest other aspects of the observation system that may change over time, we can try to respond to these in more detail.

30

References

Brandt M, Rasmussen K, Peñuelas J et al. (2017) Human population growth offsets climate-driven increase in woody vegetation in sub-Saharan Africa. *Nature Ecology & Evolution*, **1**, 81.

Carreiras J, Melo J, Vasconcelos M (2013) Estimating the Above-Ground Biomass in Miombo Savanna Woodlands (Mozambique, East Africa) Using L-Band Synthetic Aperture Radar Data. *Remote Sensing*, **5**, 1524–1548.

Chave J, Condit R, Aguilar S, Hernandez A, Lao S, Perez R (2004) Error propagation and
40 scaling for tropical forest biomass estimates. *Phil. Trans. R. Soc. B*, **359**, 409–420.

- Chave J, Réjou-Méchain M, Búrquez A et al. (2014) Improved allometric models to estimate the aboveground biomass of tropical trees. *Global Change Biology*, n/a-n/a.
- Frost P (1996) The ecology of Miombo woodlands. In: *The Miombo in transition: woodlands and welfare in Africa* (ed Campbell B), pp. 11–55. CIFOR, Bogor, Indonesia.
- Hill TC, Williams M, Bloom a A, Mitchard ET a, Ryan CM (2013) Are inventory based and remotely sensed above-ground biomass estimates consistent? *PloS one*, **8**, e74170.
- Liu YY, van Dijk AIJM, de Jeu R a M, Canadell JG, McCabe MF, Evans JP, Wang G (2015) Recent reversal in loss of global terrestrial biomass. *Nature Climate Change*, **5**, 1–5.
- 10 Lucas R, Armston J, Fairfax R et al. (2010) An evaluation of the ALOS PALSAR L-band backscatter - Above ground biomass relationship Queensland, Australia: Impacts of surface moisture condition and vegetation structure. *IEEE Journal of Selected Topics in Applied Earth Observations and Remote Sensing*, **3**, 576–593.
- McNicol IM, Berry NJ, Bruun TB, Hergoualc’h K, Mertz O, de Neergaard A, Ryan CM (2015) Development of allometric models for above and belowground biomass in swidden cultivation fallows of Northern Laos. *Forest Ecology and Management*, **357**, 104–116.
- Mermoz S, Le Toan T, Villard L, Réjou-Méchain M, Seifert-Granzin J (2014) Biomass assessment in the Cameroon savanna using ALOS PALSAR data. *Remote Sensing of Environment*, **155**, 109–119.
- 20 Mitchard ET a, Flintrop CM (2013) Woody encroachment and forest degradation in sub-Saharan Africa’s woodlands and savannas 1982-2006. *Philosophical Transactions of the Royal Society B: Biological Sciences*, **368**, 20120406.
- Mitchard ETA, Saatchi SS, Woodhouse IH et al. (2009) Using satellite radar backscatter to predict above-ground woody biomass: A consistent relationship across four different African landscapes. *Geophysical Research Letters*, **36**.
- Mitchard ET a., Saatchi SS, Lewis SL et al. (2011) Measuring biomass changes due to woody encroachment and deforestation/degradation in a forest–savanna boundary region of central Africa using multi-temporal L-band radar backscatter. *Remote Sensing of Environment*, **115**, 2861–2873.
- 30 Mitchard ET, Saatchi SS, Baccini A, Asner GP, Goetz SJ, Harris NL, Brown S (2013) Uncertainty in the spatial distribution of tropical forest biomass: a comparison of pan-tropical maps. *Carbon balance and management*, **8**, 10.
- Mugasha WA, Eid T, Bollandsås OM, Malimbwi RE, Chamshama SAO, Zahabu E, Katani JZ (2013) Allometric models for prediction of above- and belowground biomass of trees in the miombo woodlands of Tanzania. *Forest Ecology and Management*, **310**, 87–101.
- Nzunda EF, Griffiths ME, Lawes MJ (2014) Resource allocation and storage relative to resprouting ability in wind disturbed coastal forest trees. *Evolutionary Ecology*, **28**, 735–

- Oliver C, Quegan S (1998) *Understanding Synthetic Aperture Radar Images*. 479 pp.
- Poulter B, Frank D, Ciais P et al. (2014) Contribution of semi-arid ecosystems to interannual variability of the global carbon cycle. *Nature*, **509**, 600–603.
- Reiche J, Lucas R, Mitchell AL et al. (2016) Combining satellite data for better tropical forest monitoring. *Nature Climate Change*, **6**, 120–122.
- Ryan CM (2009) *Carbon cycling, fire and phenology in a tropical savanna woodland in Nhambita, Mozambique*. PhD Thesis, University of Edinburgh, Vol. PhD. University of Edinburgh.
- 10 Ryan CM, Hill T, Woollen E et al. (2012) Quantifying small-scale deforestation and forest degradation in African woodlands using radar imagery. *Global Change Biology*, **18**, 243–257.
- Ryan CM, Pritchard R, McNicol I, Lehmann C, Fisher J (2016) Ecosystem services from Southern African woodlands and their future under global change. *Philosophical transactions of the Royal Society of London. Series B, Biological sciences*.
- Ryan CM, Williams M, Grace J, Woollen E, Lehmann CER (2017) Pre-rain green-up is ubiquitous across southern tropical Africa: implications for temporal niche separation and model representation. *New Phytologist*, **213**, 625–633.
- 20 Shimada M, Isoguchi O, Tadono T, Isono K (2009) PALSAR radiometric and geometric calibration. *IEEE Transactions on Geoscience and Remote Sensing*, **47**, 3915–3932.
- Shimada M, Itoh T, Motooka T, Watanabe M, Shiraishi T, Thapa R, Lucas R (2014) New global forest / non-forest maps from ALOS PALSAR data (2007 – 2010). *Remote Sensing of Environment*, **155**, 13–31.
- Tanase MA, Panciera R, Lowell K, Tian SY, Garcia-Martin A, Walker JP (2014) Sensitivity of L-Band Radar Backscatter to Forest Biomass in Semiarid Environments: A Comparative Analysis of Parametric and Nonparametric Models. *Ieee Transactions on Geoscience and Remote Sensing*, **52**, 4671–4685.
- Le Toan T, Quegan S, Davidson MWJ et al. (2011) The BIOMASS mission: Mapping global forest biomass to better understand the terrestrial carbon cycle. *Remote Sensing of Environment*, **115**, 2850–2860.
- 30 Williams M, Ryan CMM, Rees RMM et al. (2008) Carbon sequestration and biodiversity of re-growing miombo woodlands in Mozambique. *Forest Ecology and Management*, **254**, 145–155.
- Woodhouse IH, Mitchard ET a, Brolly M, Maniatis D, Ryan CM (2012) Radar backscatter is not a “direct measure” of forest biomass. *Nature Climate Change*, **2**, 556–557.

Reviewers' comments:

Reviewer #1 (Remarks to the Author):

I found the response to the referees's concerns useful and satisfactory and I liked the newly-developed probabilistic approach.

My only remaining reservation remains the temporal stability of the signal that may be significant in the region due to climatic contingencies (even if the authors illustrate the global stability of ALOS through the work done by Shimada et al.). This point has been mostly raised by referee 3 and I believe that climatic data, from local meteorological stations or from satellite, may be used to assess any concomitant variation between backscatters and climate variability in the region. This would help to rule out or not the hypothesis that different moisture conditions may have influenced your results.

Otherwise, I still believe that this paper constitutes an important step toward the recognition of the importance of drylands in the C cycle and will probably inspire future Radar works with its original approach accounting for errors and reporting uncertainties associated with LCC.

Minor:

Comment 22: RMSE is not only a random error, it also integrates the bias in its measurement (i.e. a pure systematic error, with no random effect, will produce a non-zero RMSE).

Line 72: Just a thought: A FAO paper published during the present paper revision may be eventually discussed (Bastin et al. 2017 Science).

Line 79: Missing square bracket.

Line 286: replace "in" to "if"?!

Reviewer #2 (Remarks to the Author):

I have read the new version of the manuscript entitled "Carbon losses from deforestation and widespread degradation offset by extensive growth in African woodlands" by Iain M. McNicol, Casey M. Ryan, and Edward T.A. Mitchard. Authors have made efforts to explain better their methodological approach and to answer in details to reviewers comments. Nonetheless, before the manuscript could be accepted for publication, I would have some comments to make, especially regarding the probabilistic approach. These comments could potentially change the results of the study (number reported) but should a priori not change the main conclusions of the paper, which deserve to be published.

In response to my third and most important comment (numbered [22] in the list of comments from the three reviewers), authors argue that RMSE computed at the pixel scale cancel out for large areas. I agree, this is why you can use a biomass-backscatter relationship to estimate carbon stock at one date on large areas. You end up with a small uncertainty (the total RMSE for the area is given as $\sqrt{\text{RMSE}^2 \cdot n}$). But this has nothing to do with the fact that you can estimate small carbon stock changes at the pixel scale. For that, the probabilistic approach used by the authors seems to be an appropriate solution.

The probabilistic approach is better explained than in the first version of the manuscript, especially thanks to the addition of Fig. S4 and part 2.5.1 in the Supplementary Materials. Thanks to the authors for their efforts to explain better this step which is to my point of view, the most important

and original point in their analysis.

Nonetheless more information is necessary to explain better how the estimated changes in each land cover change category is summed-up at the landscape level (see. l. 323 of the SI): "The area affected by each LCC was obtained by summing the relevant probabilities over the area of interest". Please be more specific.

Also, authors underlined that "The probability that the four land cover changes have occurred do not sum to unity" (l. 324 in SI). Why was it not possible to build a probability approach with probabilities summing to one?

I think that some probability definitions might have to be re-written:

In the gain process, why not considering the case where $P(B07 < 10)$? I would have written $P(\text{gain}) = P(B10 \geq 10) P(B10 > B07)$

In the degradation process, the case where $B07 = 10$ should be excluded, so I would have written $P(\text{degradation}) = P(B07 > 10) P(B10 \geq 10) P([B10 / B07] < 0.8)$

In the minor losses process, why the condition $P(B07 \geq 10)$ is not considered ? To my point of view, it should be $P(\text{minor losses}) = P(B07 \geq 10) P(B07 > B10) P([B10 / B07] \geq 0.8)$

The deforestation process seems OK: $P(\text{deforested}) = P(B07 \geq 10) P(B10 < 10) P([B10 / B07] < 0.8)$

I would have added an additional process to make probabilities sum-up to one: $P(\text{not forest}) = P(B07 < 10) P(B10 < 10)$.

I might be wrong with the suggestions above but probability definitions must be better argued.

The manuscript is missing some informations about the scientific softwares and libraries used for the analysis which are computationally demanding. Some words on that should help researchers to reproduce the results of the study in the future. Adding some piece of code for a representative example on a repository like GitHub could also help.

I hope these comments will help improve the clarity of the manuscript and increase the impact of the article.

Reviewer #3 (Remarks to the Author):

Concerning 'temporal constancy', I agree that long time series of biophysical measures can be used for change detection, given a demonstrated consistent calibration over the data record. However, bi-temporal and other temporally limited approaches do not meet this requirement. If ephemeral effects of soil moisture impacted your analysis, a 30 year run of your model could detect such variations against a backdrop of monotonic biomass accumulation in areas not experiencing disturbance. Likening your study to ones that employ decade+ time-series is not appropriate.

'We also note that the relationship is about as good as it gets for calibration of biomass-backscatter curves, which vary from 0.27 to 0.86.' This is not an argument for stating that miombo woodlands are accumulating carbon sufficient to offset emissions due to land change. For example, if two land classifications, each with an accuracy of 95% are used to quantify change, ~10% of mapped change will in fact be error due to convolved errors of the individual maps. Employing a highly uncertain regression model in a similar fashion must convolve error, and the

response to attribute this error source to speckle is not reasonable. The model is not robust enough to map change without removing expected or validated commission errors.

'we have taken considerable care and effort to mask out areas that might be wetlands.' I disagree with this statement, as it reflects a degree of geographic ignorance. Wetlands take many forms in Zambezian savannas – for example dambos are found throughout the study area and are not attached to open water bodies. The idea that a buffer around water bodies solves the wetland issue is overly simplistic.

'the vast majority of the imagery used in the mosaic comes from the mid to late dry season where the observations are relatively insensitive to difference in precipitation' Do you mean to say there is no precipitation? This is the bet made with sparse data collections – the assumption that if we acquire an image in a dry season window, we can ignore signals that might confound or impact our model. Dense time-series can overcome this, as mentioned before. Also, there are interannual climate variations in tropical dry savannas that result in canopy cover change. Again, a dense time-series would also likely overcome such interannual variation.

In summary, I do not accept the findings of this study – there are too many assumptions required to accept the results. The limited time-series inputs, short study period interval, and change detection method all raise serious concerns about the validity of the study.

**Overview**

We would like to thank the reviewers for their second round of helpful and insightful
comments on the manuscript. We are pleased that two of the reviewers were satisfied with the
majority of our replies, and that they feel that our paper is worthy of publication in Nature
Communications.

We have done our best to constructively engage with Reviewer 3, but are of course
disappointed that she/he states that they do not accept the findings of the study. We note the
R3 has not engaged with any of the radar physics that supports the assumptions that underpin
our method, and which have passed peer-review many times before (Mitchard *et al.*, 2011,
2013, Ryan *et al.*, 2012, 2014; Collins & Mitchard, 2015; Joshi *et al.*, 2015). We would of
course welcome further specific comments about how the manuscript could be improved, but
most of the comments seem to be more to do with general criticism of wide-area remote
sensing and the lack of a dense time series. As there are not data to allow a dense time series
approach as advocated by R3, it is not clear to us that we can address the remaining
comments. We would also like to point out that our results, particularly the location of the
observed positive and negative changes, are similar to many existing studies, all of which use
different methods to the ones used here (Mitchard & Flintrop, 2013; Baccini *et al.*, 2017;
Brandt *et al.*, 2017).

As with our 1st set of responses, we first describe the major changes we have made in
response to the reviewer's comments; second, we provide a detailed line by line response to
the remaining queries.

**Summary of major changes**

**Soil moisture effects now more fully accounted for (Reviewers 1 & 3)**

**Reviewer 1**

[01] My only remaining reservation remains the temporal stability of the signal that may
be significant in the region due to climatic contingencies (even if the authors illustrates the
global stability of ALOS through the work done by Shimada et al.). This point has been
mostly raised by referee 3 and I believe that climatic data, from local meteo stations or from
satellite, may be used to assess any concomitant variation between backscatters and climate
variability in the region. This would help to rule out or not the hypothesis that different
moisture conditions may have influence your results.

To resolve this issue, which is related to some (but by no means all) of R3's criticisms, we
undertook an analysis of the influence of soil moisture on our results. To do this we looked at
the change in biomass as a function of the soil moisture change between the 2007 and 2010
observations (Fig R1). This showed a small tendency for losses to be more frequently
observed when soil moisture had decreased (grey lines in Fig R1) – however the effect is
small, with even large changes in soil moisture ($\pm 40\%$ points) leading to on average a ± 1.5
MgC ha^{-1} error in the biomass change estimation, and consequently only minor changes to
the estimated area of land cover changes (Table R1).

**Figure R1** – Estimated biomass change as a function of soil moisture, before (grey; $y =$
 $0.0134x + 0.58$, $r^2 = 0.38$, $p < 0.05$) and after (black; $y = -0.004x + 0.32$, $r^2 = 0.04$ $p = 0.04$)
 the new correction has been applied. Only soil moisture changes of $\pm 40\%$ points are shown,
 as areas with larger changes are masked from the analysis.

Despite the small magnitudes involved and for completeness sake, we have implemented a
 correction to remove the effect of changes in soil moisture, based on an statistical model
 consistent with the Water Cloud Model (Attema & Ulaby, 1978). This resulted in the effect
 of soil moisture changes being almost fully removed from the land cover change estimates
 (black line in figure RR1 shows no strong trend, which incidentally now has a negative sign).
 This correction is now described in the SM, and mentioned in the main methods:

Text in main methods and SM:

*To account for seasonal moisture effects, we developed a statistical model that*
 *applies a small correction to the backscatter data in areas where the estimated soil*
 *moisture varied between years. The model provides for differential corrections*
 *according to the estimated woody biomass, following the logic of the Water Cloud*
 *Model (Attema and Ulaby 1978).*

Additional text in SM:

*It was constructed by parameterising the following regression model in R statistical*
 *software:*

$\Delta\gamma_0 \sim \Delta\theta + \Delta\theta * B07$ Eq.15

Where $\Delta\gamma_0$ is the change in radar backscatter between 2007 and 2010, $\Delta\theta$ is the
associated change in soil moisture (%), and B07 is the backscatter in 2007.”

[...]

“The effect of changes in soil moisture on estimated biomass changes is visualised
in Figure S6. The comparison revealed that reductions in backscatter tended to be
more frequently observed in areas where soil moisture decreased between 2007 and
2010, with increases more prevalent in areas where the soil moisture was greater in
2010, although the magnitude of the effect tends to be small (~ 1 MgC ha⁻¹), with the
trend diminishing in areas with higher AGC densities in 2007, as predicted by the
WCM. Extreme changes in moisture (e.g. > 40% points) resulted in some large,
idiosyncratic changes in backscatter, most notably in Central Angola where there
was some notable striping present in the data, although such areas comprise only
5% of the dataset. These areas were therefore excluded from both the analysis (5%
of the study region), and the statistical model, which was fit using only the data
where moisture varied by less than this value. The resultant model was used to apply
a small correction to the backscatter maps from 2008 – 2010, with the size of the
adjustment decreasing as AGC stocks increase (Figure S7).”

**Table R1** – Comparison of key change statistics before, and after the soil moisture
correction. We include the errors on the rates of change to show that the small difference pre-
and post-correction is well within the errors on all estimates.

	Deforested	Degraded	Minor losses	Gains
Pre-correction	7.7% [5.8 – 9.2%]	16.1% [13.2– 18.6%]	26.3% [21.4 – 32.7%]	49.8% [43 – 58%]
Post-correction	8.4% [6.3 – 9.9%]	17.0% [14.0 – 19.7%]	26.9% [22 – 33%]	47.7% [41 – 55%]

This new correction sits alongside our three existing strategies to minimise soil moisture
effects: 1) the masking of areas that have experienced large changes in soil moisture between
the 2007 and 10 observations, 2) the masking of areas that land cover maps suggest are likely
to be flooded – with an buffer around them in case of inaccuracies in the land cover products
and 3) the exclusion of low biomass areas from the land cover change statistics, as these areas
are where the soil moisture signal is strongest.

**New parameterisation of the gamma distribution for probabilistic change**
**detection (R2)**

Following comments from R2 [paragraph 14 in the first review response] regarding our
probability matrices, we have made two changes to the way that the gamma distributions of
backscatter are modelled. Firstly, we have increased the number of pseudo-homogenous areas
that were used to parameterise the distributions (from 11 to 20 including more areas at the top

and bottom of the backscatter gradient). Secondly, we now allow k to vary, rather than
assuming it is fixed across the mosaic. This allows a better fit to the observed data,
particularly in the lower backscatter regions. This change is now documented in the SM
methods section 2.5.

*“The shape parameter, k , is effectively the equivalent number of looks in a PALSAR*
*25 m mosaic, and was calculated by dividing the observed, or expected backscatter*
*value, by the scale parameter.”*

With this new approach, a pixel that, for example, loses >20% of its original value, and
moves just below our 10 MgC ha⁻¹ wooded/ non-wooded threshold, is now correctly assigned
a higher probability of deforestation rather than to degradation. As a result of this, we now
find that our deforestation area is now 5x higher than Hansen et al. (2013). The differences
can be explained by the differing definitions between the two studies with Hansen et al.
(2013) only measuring areas that were completely cleared, with such areas only comprising
10% of our data. Our approach now also detects more of the deforestation (60%) detected in
Hansen, with the remaining conclusions unaffected.

**Response to Reviewer 1**

[02] I found the response to the referee’s concerns useful and satisfactory and I liked the
newly-developed probabilistic approach.

[03] My only remaining reservation remains the temporal stability of the signal that may
be significant in the region due to climatic contingencies (even if the authors illustrates the
global stability of ALOS through the work done by Shimada et al.). This point has been
mostly raised by referee 3 and I believe that climatic data, from local meteo stations or from
satellite, may be used to assess any concomitant variation between backscatters and climate
variability in the region. This would help to rule out or not the hypothesis that different
moisture conditions may have influence your results.

Thank you for this suggestion. See major changes above for our new method to address this,
based on this suggestion.

[04] I still believe that this paper constitutes an important step toward the recognition of
the importance of drylands in the C cycle and will probably inspire future Radar works with
its original approach accounting for errors and reporting uncertainties associated with LCC.

We thank the reviewer for their comments on our manuscript.

[05] Minor:

Comment 22: RMSE is not only a random error, it also integrates the bias in its measurement
(i.e. a pure systematic error, with no random effect, will produce a non-zero RMSE).

Yes, you are of course correct

[06] Line 72: Just a thought: A FAO paper published during the present paper revision
may be eventually discussed (Bastin et al. 2017 Science).

We were pleased to see this paper and its emphasis on the importance of trees in drylands.
However, the Bastin et al analysis is restricted to drylands, which only cover a small

proportion of our study region, it is not possible to directly compare their results to our data,
as we have done with the FAO statistics on Line 72.

[07] Line 79: Missing square bracket.

[08] Line 286: replace “in” to “if”?!

Text has been changed accordingly for both.

**Response to Reviewer 2**

[09] I have read the new version of the manuscript untitled "Carbon losses from
deforestation and widespread degradation offset by extensive growth in African woodlands"
by Iain M. McNicol, Casey M. Ryan, and Edward T.A. Mitchard. Authors have made efforts
to explain better their methodological approach and to answer in details to reviewers
comments. Nonetheless, before the manuscript could be accepted for publication, I would
have some comments to make, especially regarding the probabilistic approach. These
comments could potentially change the results of the study (number reported) but should a
priori not change the main conclusions of the paper, which deserve to be published.

Thanks for your suggestions below, and for emphasising the importance of the main
conclusions.

[10] In response to my third and most important comment (numbered [22] in the list of
comments from the three reviewers), authors argue that RMSE computed at the pixel scale
cancel out for large areas. I agree, this is why you can use a biomass-backscatter relationship
to estimate carbon stock at one date on large areas. You end up with a small uncertainty (the
total RMSE for the area is given as $\sqrt{\text{RMSE}^2 \cdot n}$). But this has nothing to do with the fact
that you can estimate small carbon stock changes at the pixel scale. For that, the probabilistic
approach used by the authors seems to be an appropriate solution.

Thank you – it is helpful to separate these two in this discussion.

[11] The probabilistic approach is better explained than in the first version of the
manuscript, especially thanks to the addition of Fig. S4 and part 2.5.1 in the Supplementary
Materials. Thanks to the authors for their efforts to explain better this step which is to my
point of view, the most important and original point in their analysis.

Nonetheless more information is necessary to explain better how the estimated changes in
each land cover change category is summed-up at the landscape level (see. l. 323 of the SI):
"The area affected by each LCC was obtained by summing the relevant probabilities over the
area of interest". Please be more specific.

Yes, we could have been much clearer here. We have added a section to the SM, which
describes this mathematically:

*“The area affected by each LCC, and the total wooded and non-wooded area, was*
*obtained by summing the relevant probabilities, or weighted estimates over the area*
*of interest as follows:*

$$38 \quad \sum_{i=1}^n P(\text{wooded})_{i\dots n} \quad \text{Eq.10}$$

$$39 \quad \sum_{i=1}^n P(\text{deforestation})_{i\dots n} \quad \text{Eq.11}$$

$\sum_{i=1}^n P(\text{degradation})_{i\dots n}$ Eq.12

$\sum_{i=1}^n P(\text{gain})_{i\dots n}$ Eq.13

$\sum_{i=1}^n P(\text{minor loss})_{i\dots n}$ Eq.14

*Where n is the number of pixels within the area of interest.”*

We hope this clarifies the approach sufficiently.

[12] Also, authors underlined that "The probability that the four land cover changes
[LCCs] have occurred do not sum to unity" (l. 324 in SI). Why was it not possible to build a
probability approach with probabilities summing to one?

The reason that the 4 LCCs do not sum to 1, is that there is a probability that the pixel was
not ≥ 10 MgC ha⁻¹ in 2007 and thus was not included in the analysis of “wooded land”. If you
include the probability that the pixel is < 10 MgC ha⁻¹ the probabilities do sum to 1. To make
this all clearer we have added the following to the SM:

*“There is also the probability that the pixel was non-wooded in 2007:*

$P(\text{non-wooded}) = P(B_{07} < 10)$ Eq.7”

[13] I think that some probability definitions might have to be re-written:

- • In the gain process, why not considering the case where $P(B_{07} < 10)$? I would have
written $P(\text{gain}) = P(B_{10} \geq 10) P(B_{10} > B_{07})$

Our definition of wooded land is that it has $B_{07} \geq 10$, so to get the gain area of wooded land we
need to retain this term. We do not account for any LCCs in areas with $B_{07} < 10$.

- • In the degradation process, the case where $B_{07} = 10$ should be excluded, so I would
have written $P(\text{degradation}) = P(B_{07} > 10) P(B_{10} \geq 10) P([B_{10} / B_{07}] < 0.8)$

Thank you. The inclusion of the “=” sign in argument “ $B_{07} \geq 10$ ” was a typo and not
included in the code that creates the probability matrices

- • In the minor losses process, why the condition $P(B_{07} \geq 10)$ is not considered ? To
my point of view, it should be $P(\text{minor losses}) = P(B_{07} \geq 10) P(B_{07} > B_{10}) P([B_{10}$
$/ B_{07}] \geq 0.8)$

Our code and definition did include the term $P(B_{07} \geq 10)$ but was not included in text.
We’ve added this as per your suggestion.

- • The deforestation process seems OK: $P(\text{deforested}) = P(B_{07} \geq 10) P(B_{10} < 10)$
$P([B_{10} / B_{07}] < 0.8)$

- • I would have added an additional process to make probabilities sum-up to one: $P(\text{not}$
$\text{forest}) = P(B_{07} < 10) P(B_{10} < 10)$.

Done, see above

I might be wrong with the suggestions above but probability definitions must be better
argued.

Thanks for this. Hopefully the changes above have cleared this up.

[14] The manuscript is missing some information about the scientific software and
libraries used for the analysis which are computationally demanding. Some words on that
should help researchers to reproduce the results of the study in the future.

We have included the following text at the start of the Supplementary methods:

*“We used the Geospatial Data Abstraction Library (GDAL, 2017), implemented*
*using the Python programming language version 3 (Python Software Foundation,*
<https://www.python.org/>*), for all of our data processing and analysis. All statistical*
*analyses and Figures were created using R Statistical Software (R Core Team,*
*2014).”*

[15] Adding some piece of code for a representative example on a repository like GitHub
could also help.

We will make our data and documentation and some code available on the web as soon as the
paper is published.

**Response to reviewer 3**

[16] Concerning ‘temporal constancy’, I agree that long time series of biophysical
measures can be used for change detection, given a demonstrated consistent calibration over
the data record. However, bi-temporal and other temporally limited approaches do not meet
this requirement. If ephemeral effects of soil moisture impacted your analysis, a 30 year run
of your model could detect such variations against a backdrop of monotonic biomass
accumulation in areas not experiencing disturbance. Likening your study to ones that employ
decade+ time-series is not appropriate.

We have outlined the new correction for soil moisture, and the existing methods we used to
minimise its influence above. We respond to the suggestion that canopy phenology might
influence our results below. If there are other specific assumptions that reviewer 3 feels are
unwarranted, or need more empirical support, we would be happy to address them.

[17] ‘We also note that the relationship is about as good as it gets for calibration of
biomass-backscatter curves, which vary from 0.27 to 0.86.’ This is not an argument for
stating that miombo woodlands are accumulating carbon sufficient to offset emissions due to
land change. For example, if two land classifications, each with an accuracy of 95% are used
to quantify change, ~10% of mapped change will in fact be error due to convolved errors of
the individual maps. Employing a highly uncertain regression model in a similar fashion must
convolve error, and the response to attribute this error source to speckle is not reasonable.
The model is not robust enough to map change without removing expected or validated
commission errors.

As stated both in our previous response, and in the methods section of the main text (see
quote), our uncertainty estimation takes the convolved errors into account:

*“Uncertainties on all quantities were estimated through the propagation of the*
*uncertainty in the biomass-backscatter relationship, including the bias. We*
*employed a 5000 x 2-fold cross-validation procedure, withholding half of the ground*

*data used to calibrate the radar data, and using the remainder to generate the*
*biomass-backscatter relationship. This uncertainty procedure was applied to a*
*random subsample of 5% the study area, comprising 2000 x 100 km² areas*
*randomly distributed across the study area. For each area, we calculated all derived*
*quantities, retaining the 2.5th and 97.5th percentiles of the 5000 estimates, and*
*using these 95% CI to approximate the uncertainties over the whole study area.”*

We have also shown that, as is common with remote sensing, many of the errors are
consistent over time – see Ryan et al (2012), *Glob Change Biol*. If there are specific changes
that the reviewer would like to see to the uncertainty estimation procedure, we would be
happy to investigate them.

[18] ‘we have taken considerable care and effort to mask out areas that might be
wetlands.’ I disagree with this statement, as it reflects a degree of geographic ignorance.
Wetlands take many forms in Zambebian savannas – for example dambos are found
throughout the study area and are not attached to open water bodies. The idea that a buffer
around water bodies solves the wetland issue is overly simplistic.

We would like to thank R3 for highlighting the importance of dambos, and we use this
comment to improve and clarify our description of how wet areas are dealt with in our
analysis. We should have written:

“we have taken considerable care and effort to mask out areas that might **be subject to soil**
**moisture differences between years**”, which would have more clearly included dambos.

Judging by R3’s comments, we have not been as clear as we could have been in describing
our approach to dealing with this issue. Firstly we highlight that there are two separate issues
here:

1. Areas that are likely to be wet in the dry season need to be masked from the analysis, as the
biomass-backscatter relationship will not hold. This is achieved by using the wetland land
cover maps (this included lands classified as irrigated and flooded – not just open
waterbodies and rives), and the buffer around these areas which was to account to any
excessive flooding in these areas, and any spatial errors in the land cover mask.

2. Areas that are likely to have inter-annual variation in dry season soil moisture need to be
treated carefully, as the difference in soil moisture can impact change estimates.

Smaller dambos mostly fall into the 2nd category, based on the work described in von der
Heyden (2004) and reviewed in Ryan et al (2016). In this case our approach of masking out
areas with large differences in soil moisture will address any wetlands that have large inter-
annual variability, and our soil moisture correction will account of small inter-annual
variations.

Some larger dambos will are likely to be dealt with under the methods described in point 1.

Of course, it is well known that, by definition, dambos have very little tree cover due to water
logging restricting root growth (see e.g. Woollen, Ryan *et al.*, 2012). Thus in most cases
dambo areas will not contribute to deforestation, degradation or biomass gains, as they will
have less biomass than the 10 MgC ha⁻¹ threshold. Thus dambos are unlikely to alter the
change statistics presented.

In summary, no satellite based product can claim perfection when it comes to removing all
potential sources of error. In our case, we cannot have masked out all moisture effects across
our 5 million square kilometre study area, however the combination of (1) the satellite soil

moisture product, (2) the conservative GIS-based mask, and (3) our new soil moisture
correction, will have removed the vast majority of changes resulting from soil moisture.

Finally, although we found this to be a helpful comment, we would prefer that R3 avoids
personal language and suggestions of our ignorance in her/his comments. We don't believe
that such comments help the quality of the paper or help inspire people, particularly young
scientists, to work in scientific careers.

[19] 'the vast majority of the imagery used in the mosaic comes from the mid to late dry
season where the observations are relatively insensitive to difference in precipitation' Do you
mean to say there is no precipitation?

The full quote of our response is as follows, with an underline on the section the reviewer
may have missed:

*"the vast majority of the imagery used in the mosaic comes from the mid to late dry season
where the observations are relatively insensitive to difference in precipitation in the
preceding rainy season, as the ground is very dry by then."*

So, no we do not mean to say there is no precipitation. In Ryan *et al.*, (2017) *New Phyt.* we
show that dry season rainfall is very low. *If* there is enough precipitation in the late dry
season to alter soil moisture, then this will be taken into account through the new correction
described above and the three approaches that have always been present in the method.

[20] This is the bet made with sparse data collections – the assumption that if we acquire
an image in a dry season window, we can ignore signals that might confound or impact our
model. Dense time-series can overcome this, as mentioned before.

As there are no dense time series of long wavelength radar observations, there is not much we
can do to overcome this. However we have made assumptions consistent with the physics of
radar backscatter, which we would not characterise as a "bet". There are several decades of
work to support these assumptions (see references in: Le Toan *et al.*, 1992, 2011; Oliver &
Quegan, 1998; Saatchi *et al.*, 2012).

The existing sensors that can generate dense time series (optical/infrared, and short
wavelength radar), have very weak relationships to woody biomass, and tend to convey
information on the structure of the canopy, not the woody elements. Thus we have not used
them in this study, which is focussed on woody biomass estimation.

We would also highlight that although our change detection is based on data from 2007 and
2010, we also include data on AGC stocks from 2008 and 2009 which vary by less $\pm 1\%$ from
our 2007 and 2010 estimates. If our 2007 and/or 2010 datasets were subject to extreme bias,
for example, due to seasonal moisture effects, then we would expect much more variability
between years at least one of these values to be markedly different from the rest.

In summary, we believe both the physics of radar-land surface interactions and the empirical
findings of our manuscript support the assumptions we have made.

[21] Also, there are interannual climate variations in tropical dry savannas that result in
canopy cover change. Again, a dense time-series would also likely overcome such
interannual variation.

**Figure R3** – Illustrative diagram of the interaction between the radar signal and tree canopy/
 woody elements at various wavelengths. L-band data is used in this study and is largely
 intensive to changes in leaf display (Mitchard *et al.*, 2011; Ryan *et al.*, 2012).

The major advantage of long wavelength radar, such as that which is used in this study is that
 it is insensitive to changes in small elements of the canopy such as leaves (see Figure R3).
 Thus if there is a difference in phenology between years (which is unlikely, Ryan *et al.*,
 (2017), *New Phytologist*) this will have no impact on our results as the radar sensor is
 responding largely to woody biomass, and not the ‘greenness’ or leaf phenology of the
 vegetation as the comment seems to suggest. This is one of the main reasons why radar is
 highly suitable for measuring changes in woody biomass.

The problem is whether the soil itself is excessively wet, which is more likely to be an issue
 in lower biomass areas where a larger proportion of the radar signal interacts with the ground,
 rather than the vegetation. This is discussed in the major issues section above.

[22] In summary, I do not accept the findings of this study – there are too many
 assumptions required to accept the results. The limited time-series inputs, short study period
 interval, and change detection method all raise serious concerns about the validity of the
 study.

We would like to thank the reviewer for engaging with the manuscript. We believe that we
 did not clearly explain our approach to masking and hope that our responses and the
 additional soil moisture correction have helped to allay some of these concerns. We would be
 happy to address any more specific points to address any other concerns you may have.

**References cited**

Attema EPW, Ulaby FT (1978) Vegetation modeled as a water cloud. *Radio Science*, **13**, 357.

Baccini A, Walker W, Carvalho L, Farina M, Sulla-Menashe D, Houghton RA (2017)
 Tropical forests are a net carbon source based on aboveground measurements of gain
 and loss. *Science*, **358**, 230–234.

Brandt M, Rasmussen K, Peñuelas J et al. (2017) Human population growth offsets climate-
 driven increase in woody vegetation in sub-Saharan Africa. *Nature Ecology &*

*Evolution*, **1**, 81.

Collins MB, Mitchard ETA (2015) Integrated radar and lidar analysis reveals extensive loss
of remaining intact forest on Sumatra 2007–2010. *Biogeosciences*, **12**, 6637–6653.

von der Heyden CJ (2004) The hydrology and hydrogeology of dambos: A review. *Progress*
*in Physical Geography*, **28**, 544–564.

Joshi N, Mitchard ET, Woo N et al. (2015) Mapping dynamics of deforestation and forest
degradation in tropical forests using radar satellite data. *Environmental Research*
*Letters*, **10**, 34014.

Mitchard ET a, Flintrop CM (2013) Woody encroachment and forest degradation in sub-
Saharan Africa’s woodlands and savannas 1982-2006. *Philosophical Transactions of the*
*Royal Society B: Biological Sciences*, **368**, 20120406.

Mitchard ET a., Saatchi SS, Lewis SL et al. (2011) Measuring biomass changes due to woody
encroachment and deforestation/degradation in a forest–savanna boundary region of
central Africa using multi-temporal L-band radar backscatter. *Remote Sensing of*
*Environment*, **115**, 2861–2873.

Mitchard ET a., Meir P, Ryan CM et al. (2013) A novel application of satellite radar data:
measuring carbon sequestration and detecting degradation in a community forestry
project in Mozambique. *Plant Ecology & Diversity*, **6**, 159–170.

Oliver C, Quegan S (1998) *Understanding Synthetic Aperture Radar Images*. 479 pp.

R Core Team (2014) R: A Language and Environment for Statistical Computing.

Ryan CM, Hill T, Woollen E et al. (2012) Quantifying small-scale deforestation and forest
degradation in African woodlands using radar imagery. *Global Change Biology*, **18**,
243–257.

Ryan CM, Berry NJ, Joshi N (2014) Quantifying the causes of deforestation and degradation
and creating transparent REDD+ baselines: A method and case study from central
Mozambique. *Applied Geography*, **53**, 45–54.

Ryan CM, Williams M, Grace J, Woollen E, Lehmann CER (2017) Pre-rain green-up is
ubiquitous across southern tropical Africa: implications for temporal niche separation
and model representation. *New Phytologist*, **213**, 625–633.

Saatchi S, Ulander L, Williams M, Quegan S, LeToan T, Shugart H, Chave J (2012) Forest
biomass and the science of inventory from space. *Nature Climate Change*, **2**, 826–827.

Le Toan T, Beaudoin A, Riom J, Guyon D (1992) Relating forest biomass to SAR data. *IEEE*
*Transactions on Geoscience and Remote Sensing*, **30**, 403–411.

Le Toan T, Quegan S, Davidson MWJ et al. (2011) The BIOMASS mission: Mapping global
forest biomass to better understand the terrestrial carbon cycle. *Remote Sensing of*
*Environment*, **115**, 2850–2860.

Woollen E, Ryan CM, Williams M (2012) Carbon Stocks in an African Woodland
Landscape: Spatial Distributions and Scales of Variation. *Ecosystems*, **15**, 804–818.

REVIEWERS' COMMENTS:

Reviewer #1 (Remarks to the Author):

The authors again made a detailed and well-constructed response to the referees concerns. I would like to thank them for the care and attention with which they have undertaken this response. I am in line with the response of the authors to the referee 3 criticisms. I believe that L-band radar is today the only satellite-based approach able to retrieve biomass in such ecosystem. Other remote sensing products, that may have a longer history such as landsat are not sensitive enough to biomass. If it is true that uncertainty may be associated with the use of L-band radar to infer biomass dynamics in these systems, the authors did their best to account for the main source of errors and modelled the uncertainty to assess the robustness of their result. To summarize, this work represents, in my opinion, the most up-to-date way to assess biomass change at large scale in woodlands (if we rule out multi-temporal LiDAR acquisitions which is probably logistically and financially unrealistic at such scale).

I thus still support the acceptance of this paper in Nature Communication and specifically believe that this important work will be inspiring for the radar community, especially the way the authors account for uncertainty.

Reviewer #2 (Remarks to the Author):

I congratulate the authors for the way they have answered to Reviewers' comments.

Authors have provided convincing answers to my comments regarding the probabilistic approach. The approach is consistent and clearer.

I think the article is now suitable for publication in Nature Communications. It is a very nice contribution to the study of the role of woodland in the global carbon cycle.